# Proteogenomics decodes the evolution of human ipsilateral breast cancer

Tommaso De Marchi [1✉], Paul Theodor Pyl[2], Martin Sjöström [1,3], Susanne Erika Reinsbach [4], Sebastian DiLorenzo[5], Björn Nystedt [5], Lena Tran[1], Gyula Pekar[6], Fredrik Wärnberg [7], Irma Fredriksson[8,9], Per Malmström[1,10], Mårten Fernö [1], Lars Malmström [11], Johan Malmstöm [11] & Emma Niméus [1,12✉]

Ipsilateral breast tumor recurrence (IBTR) is a clinically important event, where an isolated in-breast recurrence is a potentially curable event but associated with an increased risk of distant metastasis and breast cancer death. It remains unclear if IBTRs are associated with molecular changes that can be explored as a resource for precision medicine strategies. Here, we employed proteogenomics to analyze a cohort of 27 primary breast cancers and their matched IBTRs to define proteogenomic determinants of molecular tumor evolution. Our analyses revealed a relationship between hormonal receptors status and proliferation levels resulting in the gain of somatic mutations and copy number. This in turn re-programmed the transcriptome and proteome towards a highly replicating and genomically unstable IBTRs, possibly enhanced by *APOBEC3B*. In order to investigate the origins of IBTRs, a second analysis that included primaries with no recurrence pinpointed proliferation and immune infiltration as predictive of IBTR. In conclusion, our study shows that breast tumors evolve into different IBTRs depending on hormonal status and proliferation and that immune cell infiltration and Ki-67 are significantly elevated in primary tumors that develop IBTR. These results can serve as a starting point to explore markers to predict IBTR formation and stratify patients for adjuvant therapy.

[1] Department of Clinical Sciences Lund, Division of Oncology, Lund University, Lund, Sweden. [2] Department of Laboratory Medicine, National Bioinformatics Infrastructure Sweden, Science for Life Laboratory, Lund, Sweden. [3] Department of Radiation Oncology, University of California San Francisco, San Francisco, USA. [4] Department of Biology and Biological Engineering, National Bioinformatics Infrastructure Sweden, Science for Life Laboratory, Chalmers University of Technology, Gothenburg, Sweden. [5] National Bioinformatics Infrastructure Sweden, Uppsala University, Science for Life Laboratory, Department of Cell and Molecular Biology, Uppsala, Sweden. [6] Department of Clinical Sciences, Division of Oncology and Pathology, Lund University, Skåne University Hospital, Lund, Sweden. [7] Department of Surgery, Institute of Clinical Sciences, Sahlgrenska Academy at the University of Gothenburg, Gothenburg, Sweden. [8] Department of Molecular Medicine and Surgery, Karolinska Institutet, Stockholm, Sweden. [9] Department of Breast, Endocrine Tumors and Sarcoma, Karolinska University Hospital, Stockholm, Sweden. [10] Department of Haematology, Oncology, and Radiation Physics, Skåne University Hospital, Lund, Sweden. [11] Department of Clinical Sciences Lund, Division of Infection Medicine, Faculty of Medicine, Lund University, Lund, Sweden. [12] Department of Surgery, Skåne University Hospital, Lund, Sweden. ✉email: tommaso.de_marchi@med.lu.se; emma.nimeus@med.lu.se

Continuous improvements in breast cancer (BC) care has reduced the risk of local recurrences[1]. Still, about 4–11% of BCs develop a ipsilateral breast tumor recurrence (IBTR) within 10 years[2,3]. IBTR is a clinically important event in BC, where an isolated in-breast recurrence is a potentially curable event but associated with an increased risk of distant metastases and breast cancer death[4–9]. The time interval between IBTR and distant metastases constitutes a therapeutic window to prevent further spread. Over the course of the disease, the primary tumor (PT) evolves by clonal expansion and changes its mutational landscape. Adjuvant treatments are effective at preventing recurrent disease, but may lead to the expansion of therapy resistant clones, such as *ESR1* mutations[10,11]. To date, there has been limited efforts in characterizing changes in the molecular phenotype of IBTR and how this relates to tumor evolution and response to therapy.

The mutational repertoire and its effect on the transcriptome and proteome in primary BCs has been analyzed in numerous studies. These reports have connected key drivers and tumor subtype e.g., *TP53* and *PIK3CA* mutations with estrogen receptor (ER) negative and positive tumors, respectively, and defined how specific mutations impact prognosis[12–14], providing opportunities for patient stratification and novel therapies. However, while investigation of distant metastases is becoming more frequent, few studies have investigated the processes that lead to the development of IBTRs. Ultra-deep sequencing studies focusing on matched primary and distant recurrent tumors have shown the relevance of specific driver mutations such as *JAK2* in promoting tumor progression and proliferation, which has in turn catalyzed new avenues for therapeutic intervention by JAK-STAT pathway inhibition[15–17]. Genomic alterations occurring between primary and recurrent cancers, such as missense mutations and copy number (CN) changes, have further clarified mutational processes involved in the evolution to distant metastases, such as APOBEC-mediated mutagenesis[18].

Here, we have employed a previously developed proteogenomics workflow[19] to determine the evolution of IBTRs at the genomic, transcriptomic, and proteomic level from corresponding matched PTs to investigate paths of mutational evolution. Additionally, PTs from this set were compared with PTs that did not develop recurrent disease to define molecular and clinical features predisposing the development of IBTR. Integrated proteogenomics analyses provide additional information regarding specific pathway activation e.g., *ERBB2*, and the consequent efficacy of inhibition therapy[20], as well as an additional depth in tumor classification and biomarker selection[21–23]. Our analysis shows that the development of BC IBTRs is dependent on both hormonal receptor status of the PT, as well as changes in the DNA replication and transcription machinery in tandem with APOBEC proteins to increase genomic instability, resulting in an increased mutational load. Furthermore, the comparative analysis of PTs revealed that tumor proliferation and immune cell infiltration were enriched in tumors that develop IBTRs, which could be recapitulated by standard clinical markers obtained via immunohistochemistry.

## Results

### Proteogenomic validation of clinical and molecular characteristics.
Here we analyzed a set of 54 samples from 27 patients who developed IBTR. The 27 tumor pairs (PTs and IBTRs) were selected from a previous multi-center study that aimed to define radiosensitivity markers[24]. The paired analysis of PT and IBTR enabled a patient-centered view of the changes in recurrent tumors, measured by the changes in the genomic, transcriptomic, and proteomic landscapes. PTs and IBTRs were characterized based on ER-, PgR-, *ERBB2*-, and Ki-67-status, histological tumor grade, molecular subtype[25], and treatments (Fig. 1a, b). When comparing the PT and IBTR subsets no statistical difference in clinical and histopathological parameters was observed (Table 1 and Supplementary Data 1). The transcriptomics and proteomics data showed a high degree of concordance with IHC evaluation of key tumor markers (Fig. 1c–g). In contrast, a significant concordance for Ki-67 was observed only in IBTRs (RNA $p < 0.01$, protein $p = 0.019$). This observation may stem from heterogeneous Ki-67 expression due to clonal selection. Interestingly, 10 tumor pairs switched tumor marker status in the transition from PT to IBTR (ER: $n = 1$; PgR: $n = 6$; Ki-67: $n = 7$), which were in most cases validated by proteogenomics (RNA level: ER $n = 1/1$, PgR $n = 4/6$, Ki-67 $n = 1/7$; protein level: PgR n = 4/6, Ki-67 $n = 1/7$; Supplementary Fig. 1a, b). The weak correlation of Ki-67 status switch to transcript and protein levels might be due to the discrepancy between transcript/protein measurements when compared to immunohistochemistry, which itself depends on analytical and pre-analytical factors[26,27]. *ERBB2*/Her2 status was confirmed at the CN, RNA, and protein level (Fig. 1h–j), with no status switch between PT-IBTR pairs. Overall, these results demonstrate concordance between biomarker status and the techniques employed in this study as well as pinpointing relevant changes in tumor markers between PTs and IBTRs.

### Changes in mutational signatures between primary and recurrent tumors.
Mutational processes involved in breast cancer recurrence are often a result of homologous recombination deficiency, APOBEC-mediated mutagenesis, or age-related genome deterioration[12,18,28]. To quantify the magnitude of genomic changes between matched PTs and IBTRs, we analyzed the frequency of base transitions and transversions, and the contribution of the 30 COSMIC mutational signatures (Supplementary Fig. 2a)[29]. Two signatures displayed high contribution across PT and IBTR samples, with signature 3, enriched in cytosine transversions (possible cause: failure of DNA double-strand break repair by homologous recombination), and signature 5, enriched in cytosine and thymine substitutions (possible cause: unknown), as the most contributing in both PTs and IBTRs (Supplementary Fig. 2b). A low contribution was observed for other signatures previously associated to BC (i.e., signature 8, 13, 17, 18)[30]. Association analysis revealed a significant relationship between loss of ER expression and higher contribution of signature 3 (Supplementary Fig. 2c), which has been associated to deficient DNA repair during replication. This association suggests a link between deficient DNA repair during replication and absence of ER transcriptional activity. In the IBTR subset, signature 3 associated with loss of both ER ($p < 0.001$) and PgR ($p = 0.017$). No significant association was found between remaining signatures and clinical variables.

Next, we compared changes in contribution of the molecular signatures between matched PTs and IBTRs (Fig. 2a). This analysis showed that the contribution of signature 3 and 9 was increased in IBTRs, while the contribution of signature 1 and 5 was decreased. Signature 17 was among the top 5 changing signatures, but was discarded as the contribution was only shown in two samples (Fig. 2b). Interestingly, the increase in signature 3 was associated with absence of hormonal receptors (ER $p = 0.074$; PgR $p = 0.021$), and high tumor grade ($p = 0.047$; Fig. 2c–e). As signature 3 has been associated with failure of double strand break repair by homologous recombination, such a process might be exacerbated in tumors with high proliferation rates (ER-PgR negative and high grade).

These results show that the changes in mutational processes during IBTR formation are impacted by key tumor features such as the presence/absence of hormonal receptors.

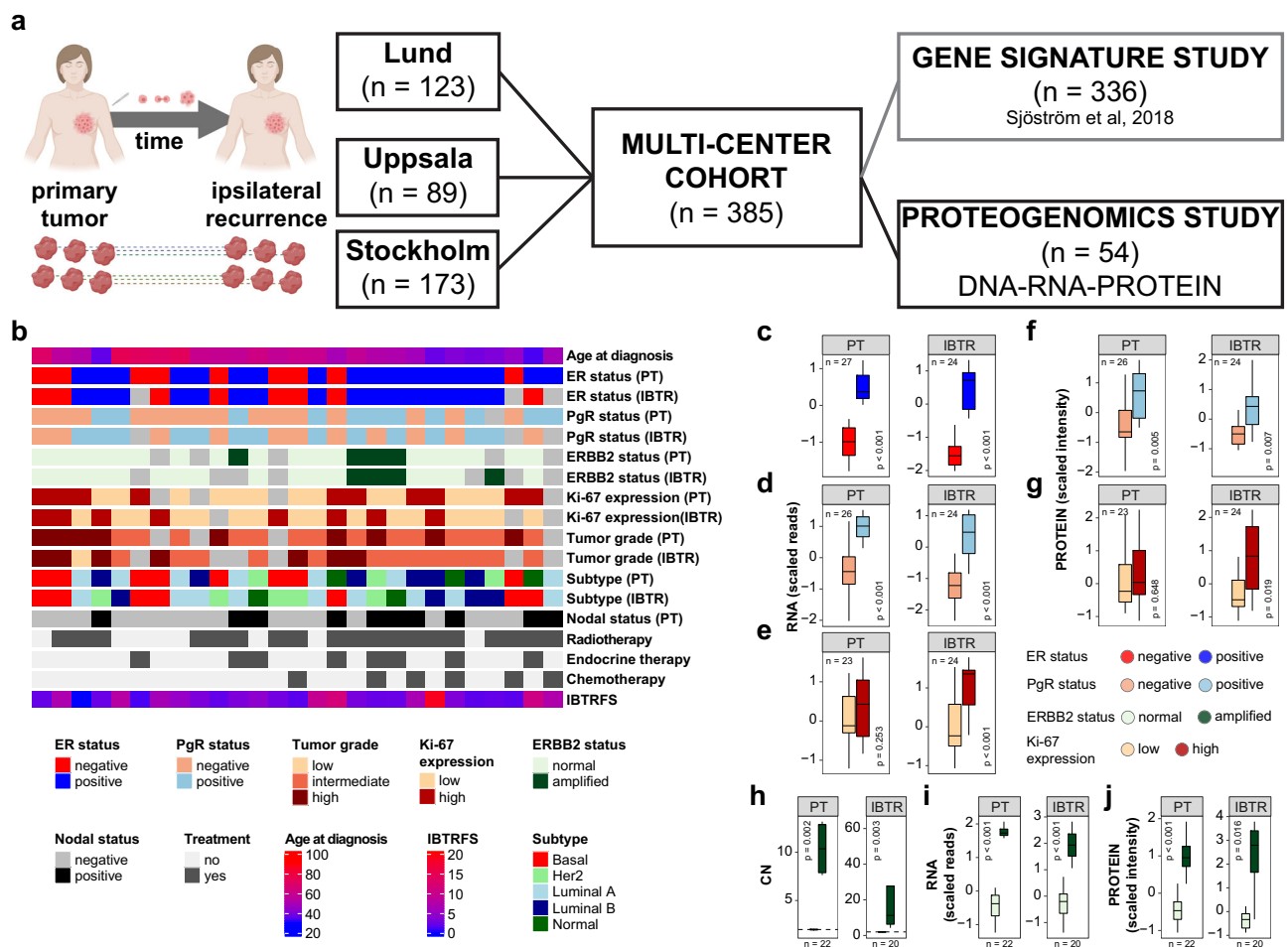

**Fig. 1 Cohort selection and metadata.** We selected a set of PTs and matched IBTRs ($n = 27 + 27$) from a larger multi-center (Lund, Uppsala, Stockholm) study (**a**; this panel was created with BioRender.com). Clinical data and histopathological characteristics were registered upon sample collection or after analyses of paraffin-embedded material, if available. **b** Description of key clinical parameters (light gray boxes represent missing values). Frozen tumors were analyzed by WGS, RNAseq, and proteomics. Transcriptomic (RNAseq, **c-e**) and proteomic (MS, **f, g**) validation of IHC-based classification of ER, PgR, and Ki-67 levels. Panels **h-j** depicts CN (WGS, left), transcript (RNAseq, center), and protein (MS, right) levels for tumors with detected ERBB2 amplification. Number of samples included in each test depends on availability of clinical data. Boxplots depict distribution of values as quartiles. Line at the center of each boxplot depict the median value. CN, copy number; ER, estrogen receptor; ERBB2/Her2, receptor tyrosine-protein kinase erbB-2; Ki-67, antigen Ki-67; IBTR, ipsilateral breast tumor recurrence; IBTRFS, IBTR-free survival; MS: mass spectrometry; PgR, progesterone receptor; PT, primary tumor.

## Copy number and mutational changes in ipsilateral breast tumor recurrences.

As we observed a significant change in the contribution of two mutational signatures between PT-IBTR pairs, we hypothesized that these events were accompanied by additional genomic changes. To address this, we analyzed the frequency of CN alterations and single nucleotide variants (SNVs). We calculated genome-wide CN changes (DeltaCN, see Methods) between PT-IBTR pairs per chromosome and characterized as either gains or losses (cutoff DeltaCN: ±0.75; median gain/sample: 363, IQR: 48.5–1115.5; median loss/sample: 95, IQR: 22–291; Supplementary Fig. 3a). Closer inspection of the top 10 CN gain and losses in each chromosome revealed that genomic regions in chromosomes 8 and 17 were frequently amplified or deleted (Supplementary Fig. 3b, c). Overall, we did not detect any association between changes in CN in the PT-IBTR pairs and CN occurrence at specific chromosomes. However, clustering of the CN changes showed a relationship between the frequency of gain/loss and hormonal receptor status (Fig. 3a). Association analysis to clinical biomarkers confirmed that absence of ER and PgR as well as high Ki-67 expression were associated with an increase in CN gains (ER $p = 0.059$, PgR $p = 0.017$, Ki-67 $p = 0.005$;

Fig. 3b–d), suggesting a relationship with lack of the ER-mediated transcriptional program and high proliferation rates. To search for molecular drivers of these relationships, we investigated whether the frequency of CN gain was linked to the expression of mutated *TP53*, which typically promotes genomic instability[31]. Despite the fact that *TP53* mutations are frequent in ER negative BCs, as also observed in our dataset ($p = 0.029$; Fig. 3e, f), no significant association between CN gains and *TP53* mutational status was observed ($p = 0.099$), suggesting other factors play a role in the establishment of CN changes in this sample set.

Next, we analyzed SNV changes for a set of previously defined key cancer genes (Nik-Zainal et al.[12]; Supplementary Fig. 4), and evaluated SNV gain/loss occurring in PT-IBTR pairs. This analysis showed that the most common SNV changes with medium or high impact were missense and stop codon gains (Supplementary Fig. 5a), with a general trend towards an increasing SNV burden in IBTR (Supplementary Fig. 5b). Further analysis showed that ER negative tumors increase in SNV gains ($p = 0.078$; Fig. 3g–j), while no significant association was observed between SNV gains and other biomarkers or clinical features. Upon assessing the most frequently mutated genes

**Table 1 Comparison between clinical and histo-pathological characteristics of primary and ipsilateral recurrent tumors.**

| | | ALL | | PRIMARY TUMOR | | IPSILATERAL BREAST TUMOR RECURRENCE | | |
| --- | --- | --- | --- | --- | --- | --- | --- | --- |
| | | *N* | % | *N* | % | *N* | % | *p*-value |
| | | 54 | 100 | 27 | 100 | 27 | 100 | |
| ER[a,b] | Positive | 34 | 63.0 | 18 | 66.7 | 16 | 59.3 | 1.000 |
| | Negative | 17 | 31.5 | 9 | 33.3 | 8 | 29.6 | |
| PgR[a,b] | Positive | 24 | 44.4 | 11 | 40.7 | 13 | 48.1 | 0.572 |
| | Negative | 26 | 48.1 | 15 | 55.6 | 11 | 40.7 | |
| Ki-67[a,b] | Low | 18 | 33.3 | 10 | 37.0 | 8 | 29.6 | 0.556 |
| | High | 29 | 53.7 | 13 | 48.1 | 16 | 59.3 | |
| ERBB2[a,b] | Normal | 34 | 63.0 | 18 | 66.7 | 16 | 59.3 | 1.000 |
| | Amplified | 8 | 14.8 | 4 | 14.8 | 4 | 14.8 | |
| Age[b] | >55 | 29 | 53.7 | 13 | 48.1 | 16 | 59.3 | 0.586 |
| | < = 55 | 25 | 46.3 | 14 | 51.9 | 11 | 40.7 | |
| Lymph-node positivity[a] | Positive | 10 | 18.5 | 10 | 37.0 | — | — | n/a |
| | Negative | 17 | 31.5 | 17 | 63.0 | — | — | |
| Grade[a,c] | Low | 1 | 1.8 | 0 | 0.0 | 1 | 3.7 | 0.549 |
| | Intermediate | 27 | 50.0 | 14 | 51.9 | 13 | 48.1 | |
| | High | 16 | 29.6 | 9 | 33.3 | 7 | 25.9 | |
| Adjuvant radiotherapy | No | 8 | 14.8 | 8 | 29.6 | — | — | |
| | Yes | 19 | 35.2 | 19 | 70.4 | — | — | |
| Adjuvant endocrine therapy | No | 19 | 35.2 | 19 | 70.4 | — | — | |
| | Yes | 8 | 14.8 | 8 | 29.6 | — | — | |
| Adjuvant chemotherapy | No | 21 | 38.9 | 21 | 77.8 | — | — | |
| | Yes | 6 | 11.1 | 6 | 22.2 | — | — | |

[a]Missing data
[b]Fisher exact test
[c]Chi-square test
*ER* estrogen receptor, *ERBB2/Her2* receptor tyrosine-protein kinase erbB-2, *Ki-67* antigen Ki-67, *PgR* progesterone receptor, *TP53* tumor protein p53.

within ER positive and negative tumors, we confirmed *PIK3CA* and *TP53* were the most commonly mutated genes in these subgroups, respectively (Supplementary Fig. 5c). These mutations were largely maintained or expanded through clonal selection in IBTRs possibly due to a conferred selective advantage towards cancer growth and survival.

Alongside CN and SNV gains, which in turn constitute a measure of tumor genomic drift and/or clonal expansion from PTs to IBTRs, we detected several losses: median SNV gain/sample = 3 (IQR: 1–4), median SNV loss/sample = 2 (IQR: 1.25–3). These likely indicate a reduction or loss of tumor sub-clones during from PT to IBTR, but did not associate with hormonal receptor status or other clinical variables with the exception of weak positive relationships with age at diagnosis of PT (Spearman Rho = 0.295) and IBTRFS (Spearman Rho = 0.372). In summary, the paired analysis conducted here suggests that primary ER and PgR negative tumors are more genomically unstable (as also reviewed in[32]), displaying a higher tendency to acquire genomic changes such as CN and SNV, resulting in highly mutated IBTRs.

**Multi-omic changes of primary breast cancer.** Having established that the absence of ER is significantly associated with the accumulation of CN in IBTRs, we investigated to what degree the genomic changes translated into alterations at the transcriptome and proteome levels.

To this end, we calculated Cosine dissimilarities between each sample within our genomic (CN), transcriptomic, and proteomic datasets (see Methods for details; Supplementary Fig. 6a–c), and extracted those between PT-IBTR pairs. A higher dissimilarity coefficient indicates more diverged IBTRs when compared to their matched PTs. We observed that dissimilarities between PT-IBTR pairs were generally greater at the RNA and protein levels when compared to CN (Supplementary Fig. 6d), indicating that

other non-genomic mechanisms contribute to the effect observed at the transcriptome and proteome level[33]. This was reflected in the distribution of dissimilarity coefficients, where RNA and protein levels showed a bi-modal trend (Supplementary Fig. 6e). As a major factor in determining accurate genomic (CN) and molecular (RNA, protein) measurements[34], tumor purity was ruled out as a potential confounder of PT-IBTR dissimilarities across all omics levels (Supplementary Fig. 6f, g).

Hierarchical clustering analysis of sample-wise CN, RNA, and protein level dissimilarities (Fig. 4a) showed that most pairs co-clustered at the CN level, while pairs were more often unmatched at the RNA and protein levels, in line with what we observed in our distribution analysis. As new PTs may be misdiagnosed as IBTRs of previous malignancies[35,36], we compared the clonal evolution of PT-IBTR pairs. Here, sample pairs with a matched normal showed an overlap between variant allele frequencies (Supplementary Fig. 7), thus indicating that IBTRs originated from their respective PTs in these patients.

Overall, we observed that the matching of PT-IBTR pairs varied in relation to data layer indicating that the changes between each tumor pair are dependent on different mechanisms, such as promoter methylation, histone binding, kinase activation, or microenvironment signaling.

To determine factors associated with IBTR formation, we assessed the relationship between dissimilarities and clinical and histo-pathological features of the cohort. Here, weak-to-moderate inverse and direct relationships were found with age at diagnosis and IBTRFS (Supplementary Fig. 8), respectively, with younger patients and fast-recurring tumors displaying a higher drift from their primaries. An inverse relationship was also observed between dissimilarity coefficients and shared PT-IBTR muta-tional load, indicating that a lower level of shared mutations between PT and IBTR is related to larger dissimilarities between

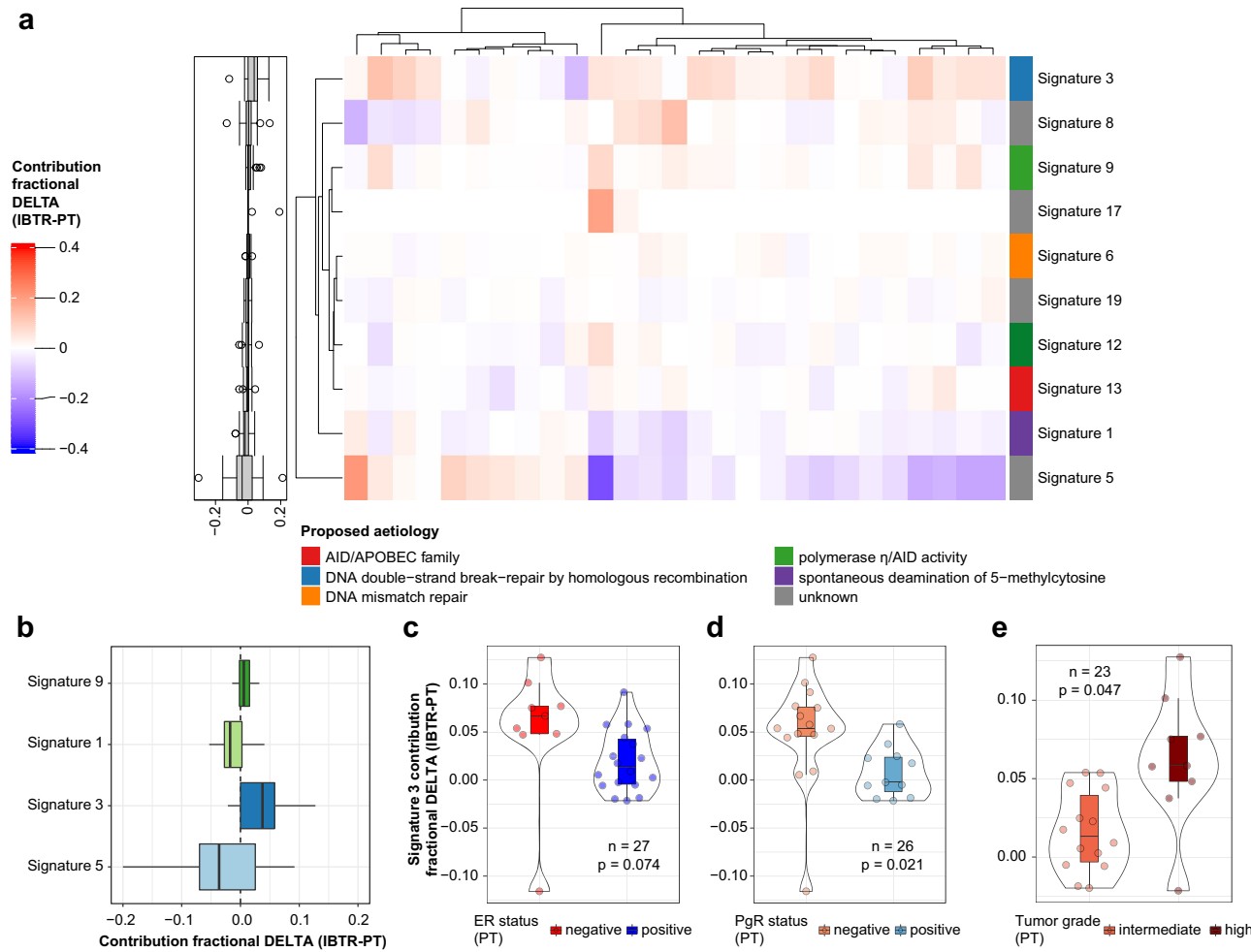

**Fig. 2 Shift of mutational signature contributions.** We evaluated the contribution of the 30 COSMIC mutational signatures within our samples (PT and IBTR subsets). Contribution fractional delta was then calculated as a measure of mutational process evolution for each PT-IBTR pair. Panel **a** displays hierarchical clustering of mutational contribution delta of the top 10 signatures with contribution gain and loss between paired samples (IBTR-PT; light gray boxes represent missing values). Boxplots left of the heatmap represent value distribution for each row. Panel **b** shows the top 5 signatures with contribution changes between primary and locally recurrent tumor pairs (excluding signature 17 due to contribution in only 2 samples). Significant associations between changes in mutational signature 3 contribution and clinical variables are depicted in **c-e**. Number of samples included in each test depends on availability of clinical data. Boxplots depict distribution of values as quartiles. Line at the center of each boxplot depict the median value. *ER* estrogen receptor, *ERBB2/Her2* receptor tyrosine-protein kinase erbB-2, *Ki-67* antigen Ki-67, *IBTR* ipsilateral breast tumor recurrence, *IBTRFS* IBTR-free survival, *PgR* progesterone receptor, *PT* primary tumor.

PT-IBTR pairs, which is turn is reflected in altered gene expression and protein abundance patterns. Overall, these results indicate that small changes at the genomic level are reflected by wider alterations at the transcript and protein ones.

In line with what be observed for mutational signatures, CN, and SNV analyses, we often observed lower PT-IBTR dissimilarities in ER positive or Luminal A tumors. On the other hand, highly proliferating tumors displayed larger dissimilarity coefficients (Fig. 4b). These results confirm our previous analyses (Figs. 2 and 3), and indicate that more substantial changes at the genomic, transcriptomic, and proteomic levels between PT-IBTR pairs might directly stem from high proliferative activity and other features typical of ER-PgR negative cancers, though no further association was detected.

**Transcriptome and proteome changes of ER positive and ER negative tumors at IBTR formation.** Having shown that ER status was associated to genomic changes between PT-IBTR pairs, we evaluated to what degree ER status impacted the

transcriptome and the proteome. Pathway analysis between ER positive and ER negative PT-IBTR pairs revealed a set of overlapping gene sets such as mTOR signaling and immune response pathways in IBTRs and PTs, respectively (Fig. 5a–d). The transition of PTs into IBTRs and the consequent changes in the mutational landscape could explain the dysregulation of proliferation-related pathways such as mTOR. Moreover, the enrichment of inflammation and immune system-related signaling in PTs might indicate changes in the relationship between the cancer and its microenvironment, possibly geared towards immune evasion. Transcript/protein pairs enriched in these pathways included inflammatory cytokines such as IL6 and IL8 as well as matrix remodeling enzymes (e.g., MMP9), indicating a tendency towards tumor invasion of the surrounding tissue.

Gene sets that showed different trends between ER positive and ER negative groups were related to pathways involved in splicing, cell cycle, and proliferation, indicating that ER negative tumors typically evolve into highly proliferative IBTRs. These were enriched in CDKs (e.g., *CDK4*) and the DNA replication machinery (e.g., *MCM3-5*) factors. We speculate that highly

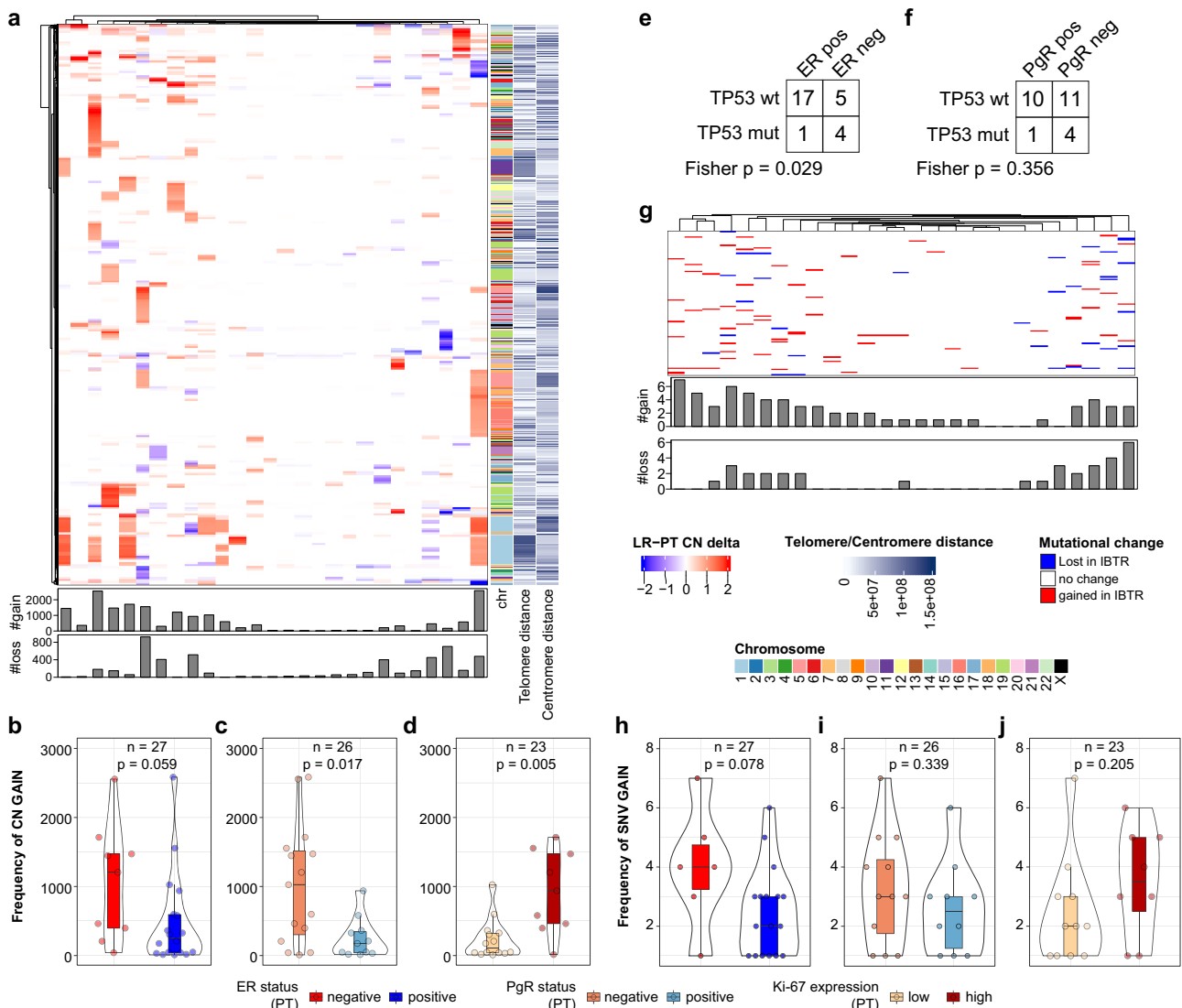

**Fig. 3 Changes in copy number and key drivers.** CN changes between paired tumors and mutational (SNV) status of COSMIC cancer genes was evaluated in our cohort. **a** Heatmap of copy number changes between primary and recurrent tumors. Bottom bar charts display sample-wise frequencies of IBTR CN gain and loss over the matched PT. **b**–**d** Association of CN gain between primary and recurrent tumors to key biomarkers. **e**, **f** Displays contingency analyses between hormonal receptor (ER and PgR) status and TP53 mutations at the PT level. **g** Association of SNV status change (i.e., gain or loss in IBTR) of cancer genes from COSMIC. **h**–**j** Association of SNV gain between primary and recurrent tumors to key biomarkers. Light gray boxes in heatmaps represent missing values. Number of samples included in each test depends on availability of clinical data. Boxplots depict distribution of values as quartiles. Line at the center of each boxplot depict the median value. Bar charts below each heatmap represent sample-wise counts for CN or SNV gains and losses. *CN* copy number, *ER* estrogen receptor, *Ki-67* antigen Ki-67, *IBTR* ipsilateral breast tumor recurrence, *IBTRFS* IBTR-free survival, *PgR* progesterone receptor, *PT* primary tumor, *SNV* single nucleotide variant, *TP53*, tumor protein p53.

proliferative IBTRs have a higher degree of replication stress, which could explain the higher mutational load in ER negative IBTRs. The increase in mutational load could further be accelerated by dysfunctional DNA repair mechanisms. This analysis was supported by analysis of transcript/protein pairs belonging to cell cycle and DNA repair terms (Gene Ontology Biological Process) showing a higher expression of these genes in IBTRs derived from ER negative PTs (Fig. 5e, f). As our previous analyses showed that *TP53* mutations was only sporadically associated to genomic, transcriptomic, or proteomic changes within our sample set, we argued that additional factors are likely involved in the accumulation of mutational features and high proliferation rates as indicated by high Ki-67 levels. The higher expression of the MYC oncogene in IBTRs derived from ER negative PTs (RNA level: fold increase 2.26, *p*-value < 0.001) is a

likely driver for the high proliferation rates. However, the accumulation of genomic features (CN and SNV) was only sporadically associated with absence of ER or high Ki-67, suggesting that other drivers were involved in the mutational changes in the IBTRs. Consequently, we investigated the APOBEC protein family, which has previously been shown to be a major mutational driver in BC[37,38] (Fig. 5g, h and Supplementary Fig. 9). Here *APOBEC3B* significantly correlated with Ki-67 levels (PT: Spearman Rho = 0.400, *p* = 0.072; IBTR: Spearman Rho = 0.674, *p* = 0.001; Fig. 5g) and was highly expressed in ER negative PTs and IBTRs (PT: Log2Ratio = 1.728, *p* = 0.007; IBTR: Log2Ratio = 2.456, *p* < 0.001; Fig. 5h). As APOBEC proteins are Cytosine deaminases[37], we expected an enrichment of C > X changes in ER negative tumors, which was confirmed by a borderline enrichment of C > T transitions in this

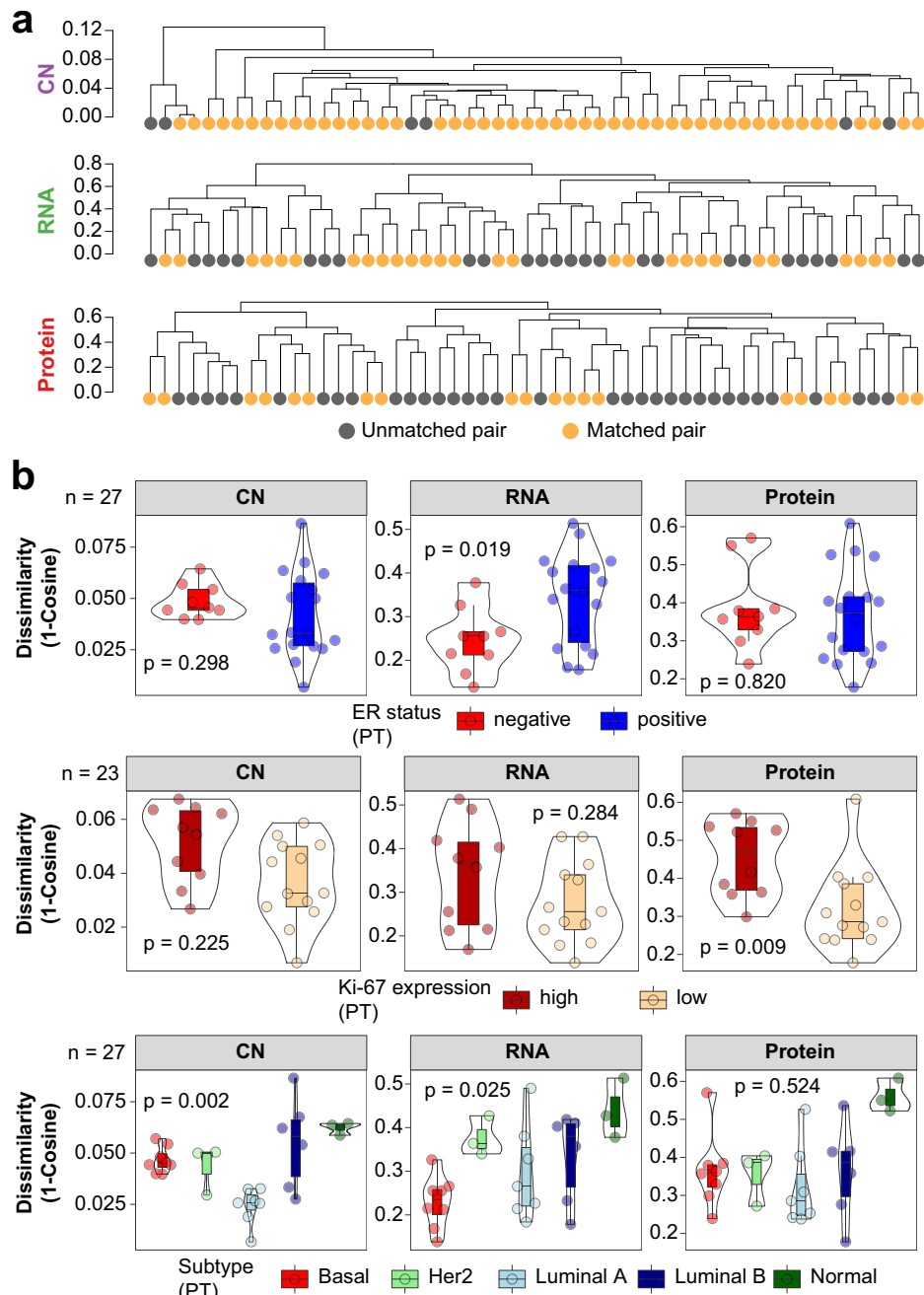

**Fig. 4 Multi-omic drift assessment of breast cancer recurrences.** Cosine dissimilarities (i.e., 1-Cosine similarity) were calculated between tumor pairs at the CN, transcript, and protein levels as a measure of evolutionary drift. **a** PT-IBTR dissimilarity-based clustering at the CN, transcriptome, and protein levels. **b** Association between clinical variables and PT-IBTR dissimilarity across data layers. Number of samples included in each test depends on availability of clinical data. Boxplots depict distribution of values as quartiles. Line at the center of each boxplot depict the median value. *CN* copy number, *ER* estrogen receptor, *IBTR* ipsilateral breast tumor recurrence, *PgR* progesterone receptor, *PT* primary tumor, *SNV* single nucleotide variant, *TP53* tumor protein p53.

group ($p = 0.076$; Fig. 5i). These results suggest that several factors work in parallel to enact the mutational and expression level drift of recurrent BCs from their PTs. These would comprise enhanced replication capacity of ER negative tumors likely driven by mechanisms outside of the ER transcriptional program as well as the expression of mutation-inducing APOBEC proteins.

**Proliferation and immune signaling are enriched in tumors that develop ipsilateral recurrences.** Having investigated the

factors that determine IBTR mutational changes and transcriptome/proteome reprogramming, we compared PT from this set (PTrec) with another set of PTs from patients that did not experience any IBTR (PTnorec; Supplementary Data 2). Comparison of clinical characteristics showed an enrichment of high grade ($p = 0.029$), high Ki-67 expression ($p < 0.001$), and higher number of lymph-node positive cases ($p = 0.003$) in the PTrec group. On top of this, no patient in the PTnorec group received adjuvant chemotherapy ($p = 0.028$; Supplementary Data 3). These differences likely stem from the fact that the PTnorec

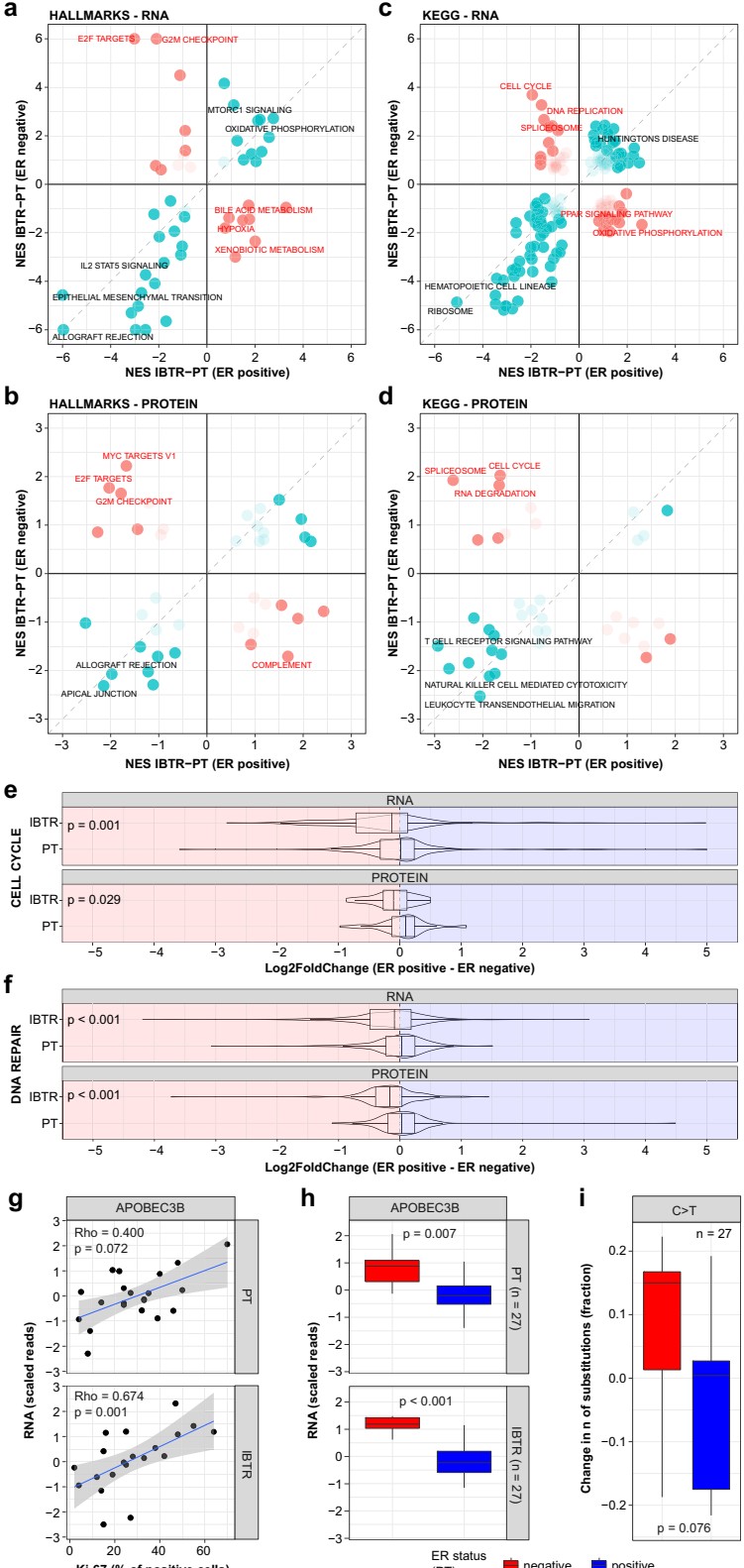

cohort did not experience any tumor relapse. Transcript and protein-level GSEA comparison of these sets revealed an enrichment of immune signaling and proliferation-related pathways in the PTrec group (Fig. 6a). GSVA scores for the two most enriched gene sets pertaining proliferation and immune signaling showed a significant relationship with IBTR-free survival (Fig. 6b, c). The GSVA scores for these pathways correlated

significantly (Fig. 6d), suggesting that increased proliferation rate and high immune cell infiltration both contribute to IBTR development. Mutations in genes involved in proliferative pathways have been associated to IBTR formation, such as PIK3CA and AKT[39]. While we observed an enrichment of such pathway in our recurring primary tumor set, there was no significant increased frequency in PIK3CA or AKT1 mutations when

**Fig. 5 Estrogen receptor expression-dependent drift of recurrent breast cancers.** Differential expression and pathway enrichment analyses were performed between paired IBTR and PT specimen within the ER positive and ER negative groups. Results were compared to measure the degree of deviation in the evolution of ER positive and ER negative tumors. **a–d** Display pathway enrichment divergence in ER positive and negative patients at the RNA and protein levels. Boxplots depicting enrichment of genes (significant and non-significant) belonging to Cell Cycle and DNA repair pathways between ER positive (blue) and ER negative (red) tumors are shown in **e** and **f**. Here changes in the distribution of Log2 fold changes were assessed by Wilcoxon test. Correlation between APOBEC3B levels and proliferation marker Ki-67 is shown in **g**. **h** Differential expression of APOBEC3B genes between ER positive and ER negative tumors. **i** Assessment of nucleotide C-to-T transition frequency changes between ER positive and negative tumors. Number of samples included in each test depends on availability of clinical data. Boxplots depict distribution of values as quartiles. Line at the center of each boxplot depict the median value. In scatter plots linear regression is depicted in blue, while gray area represents the 95% confidence interval of the regression. *ER* estrogen receptor, *Ki-67* antigen Ki-67, *IBTR* ipsilateral breast tumor recurrence, *PT* primary tumor.

compared to frequencies reported in the TCGA sample set (PIK3CA $p = 0.060$; AKT1 no mutations detected).

Next, we tested whether immune signaling and proliferation pathways could be evaluated by employing standard diagnostics, specifically Ki-67 and histological evaluation, which showed significant correlation (Spearman Rho = 0.647; Fig. 6e). Survival analysis of combined Ki-67 and infiltrating immune cell scores (expressed as percentage of cells; see Methods) displayed a borderline association (p = 0.059, Fig. 6f, g) to IBTRFS. These results suggest that high proliferating tumors and a leucocyte-rich microenvironment contribute to IBTR formation, though these findings would need verification in a larger cohort.

## Discussion

In this study we employed WGS, RNAseq, and MS-based proteomics to elucidate the processes underlying IBTR formation and to define changes in transcript/protein expression between the recurrence and its original matched PT. Overall, RNA and protein analyses corroborated the clinical markers from IHC. In several cases, there we observed receptor status switches for e.g., ER. While gain/loss of key markers is likely dependent on subclonal selection within the primary tumor[40,41], the sequencing capacity was too low to effectively reconstruct the composition and the selection of tumor sub-clones in each sample. Analysis of previously published mutational signatures[29,30] fitted onto our WGS data showed that the strongest contribution were from C > G and T > C enriched signatures in our samples, where signature 3 significantly associated to lack of ER expression. These results are consistent with previous observations in BC distant metastases[18]. In addition, signature 3 displayed the highest increase in IBTRs and was associated with ER negative tumors, which are typically characterized by a higher degree of genomic instability than ER positive cancers[42,43]. This relationship was confirmed in our CN and mutational analyses, where a higher number of CN and mutational gains was detected in the PT-IBTR ER negative pairs. Although TP53 mutations were enriched in the ER negative subset as previously reported[12], we did not observe any significant association with CN or SNV gain/loss, suggesting that other mechanisms might be driving the genomic changes in this tumor subgroup.

To assess whether the changes in genomic features impacted expression levels, paired PT-IBTR dissimilarity coefficients were calculated based on CN, RNA, and protein levels. We here found that CN drifts were smaller than the ones that impacted transcript and protein abundance. Our analyses revealed that loss of hormonal receptors and high proliferation rates not only associated with CN gains, but also with the reprogramming of both the transcriptome and the proteome. Given the fact that ER positive and negative PTs display different transcriptional programs[44,45] and often feature different sets of driver mutations (e.g., *PIK3CA vs TP53*)[12], it is reasonable to believe these features have an effect on determining the molecular and expression features of IBTRs. Differential gene/protein expression and pathway analyses

showed that ER negative IBTRs were enriched in cell cycle, DNA replication, and transcription, while ER positive tumors were geared toward metabolic pathways. ER negative breast cancer constitutes a more aggressive and recurrence-prone disease than ER positive tumors. Several studies investigated the association of ER to IBTR formation, but no difference was found to date in IBTR rates between ER positive and negative tumors[36,46,47]. In addition to cell cycle-related genes, an enrichment of *APOBEC3B* was also detected in ER negative tumors. *APOBEC3B* is a known cancer mutagen often overexpressed in BC and seemingly responsible for ~80% of the mutational load in these tumors[37,38]. APOBEC3B action in breast cancer has been shown to change in relation to the expression of ER, of which is an interactor recruited at binding sites, promoting DNA strand breaks[48]. This interaction is responsible for poor clinical outcomes in ER positive BCs[48–50]. While ER negative tumors have been reported to express high levels of APOBEC3B[49], this has not been linked to clinical outcome nor have its effect on the mutational landscape of these tumor subset been characterized. ER negative tumors are generally indicative of poor prognosis due to the fact that multiple mechanisms are enacted to enable tumor cell proliferation outside of the ER transcriptional program[12,14,45], conferring new features to cancer cells such tissue invasion[51] or immune evasion[52]. Moreover, several studies have shown that ER negative BCs constitute a molecularly heterogeneous group[53–56]. In the light of this, the role and clinical association of APOBEC family members might be either concomitant to other factors, hence the non-significant contribution of APOBEC-related signatures (i.e., signature 2 and 13) in this subset, or confounded by the other processes at work in these tumors.

Further investigation of the role of APOBEC family members in ER negative BCs would entice the analysis of subtype-stratified cohorts to better define their relationship with clinical outcomes, mutational processes, and other key factors (e.g., immune system). Mechanistic studies assessing the interaction of APOBEC enzymes with cancer drivers (e.g., MYC) or other factors active in ER negative cancers would allow to quantify the impact on these tumors´ mutational landscape and define new drug targets or alternative treatment regimens (e.g., PARP1-inhibitors)[57].

Comparative analysis of PTs with and without IBTRS revealed that immune infiltration and proliferation signaling are elevated in PTs with subsequent IBTR, which could be recapitulated by standard pathological evaluation. Immune evasion is a key mechanism in metastasis formation, which is directed by several factors intrinsic and extrinsic to the tumor, such as the secretion of immunomodulatory cytokines or the levels of tumor infiltrating immune cells (reviewed in ref. [58]). Further investigation of infiltrating immune cell type around tumors that develop IBTRs could pinpoint a targetable mechanism to prevent recurrent disease formation.

Limitations to this study include the absence of normal tissue for the majority of tumor samples analyzed by WGS, which prevented calculation of SNV-level PT-IBTR dissimilarities and

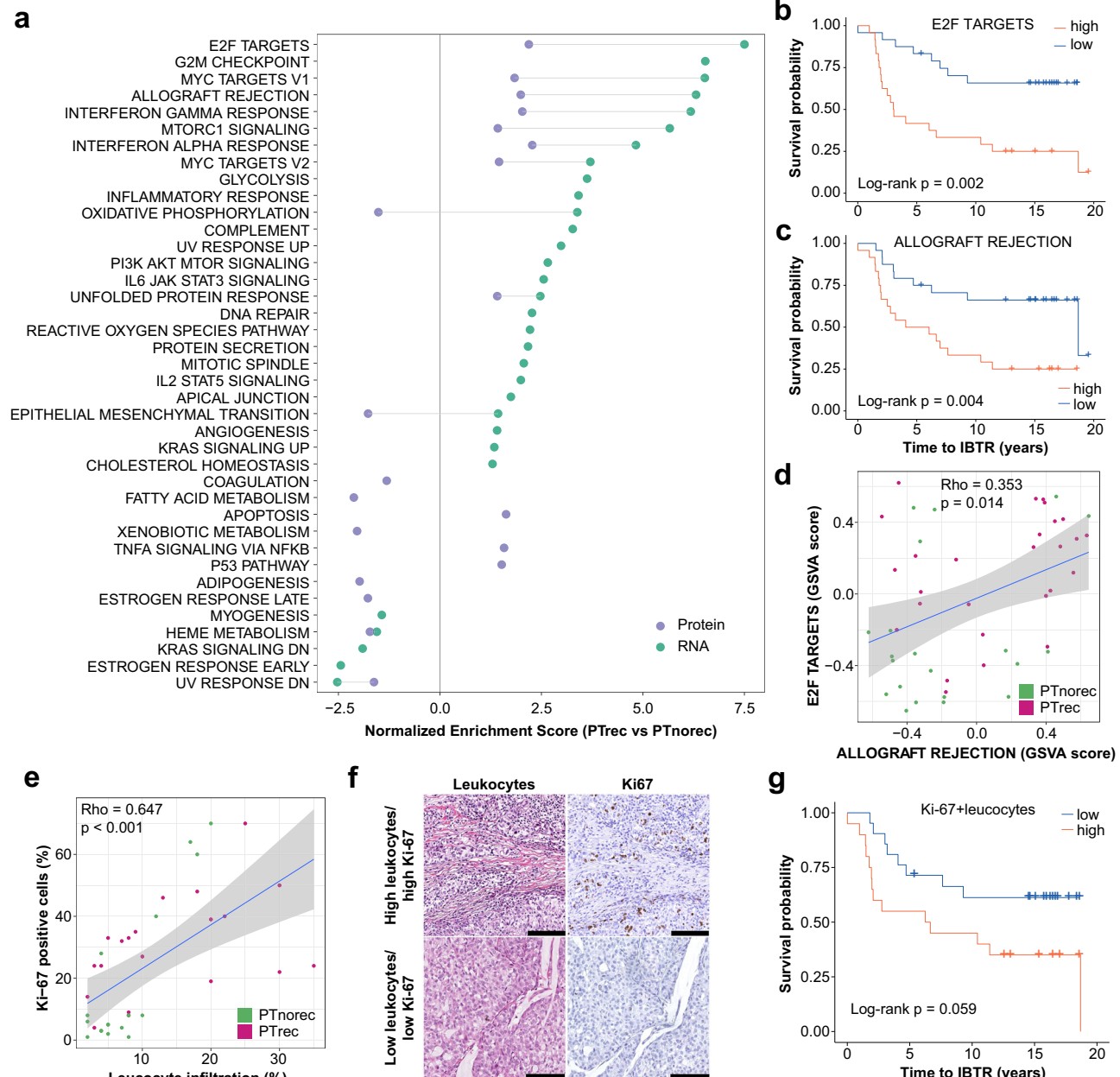

**Fig. 6 Proliferation and immune infiltration promote IBTR formation.** PTrec and PTnorec samples were compared to define possible determinants of IBTR formation. **a** GSEA analysis using HALLMARK database. **b**, **c** Survival curves of GSVA scores (cutoff: median) for top pathways pertaining to proliferation (E2F TARGETS) and immune signaling (ALLOGRAFT REJECTION). **d** Correlation analysis of GSVA scores for E2F TARGETS and ALLOGRAFT REJECTION. **e** correlation between Ki-67 and infiltrating leukocytes. **f** Examples of tissues with high and low levels of Ki-67 (evaluated by IHC) and infiltrating leukocytes (evaluated by hematoxylin-eosin staining). Black bars represent 100 μm. Panel **g** represents survival analysis for the combined Ki-67+leucocytes score (cutoff: median). Number of samples included in each test depends on availability of clinical data. In scatter plots linear regression is depicted in blue, while gray area represents the 95% confidence interval of the regression. *GSEA* gene set enrichment analysis, *GSVA* gene set variation analysis, *Ki-67* antigen Ki-67, *IBTR* ipsilateral breast tumor recurrence, *IHC* immunohistochemistry, *PT* primary tumor.

impaired variant calling and mutational signature analysis. Lack of normal tissue was due to the fact not being part of standard clinical practice at the time of collection. In addition, our study could not recapitulate IBTR features due to low power or resolve tumor clonal evolution with high resolution due to shallow sequencing. This resulted in several borderline associations that would require a larger dataset to be confirmed.

Despite of this, we here show how the mutational landscape of recurrent breast cancers diversifies based on the expression of hormonal receptors, with repercussions at the transcriptome and

proteome levels and repurposing the cell machinery towards DNA replication and proliferation, indicating these mechanisms should be targeted to prevent IBTR formation. Additionally, we elucidated the biological features of tumors that form IBTRs, which can be in turn recapitulated by standard clinical diagnostics.

## Methods

**Sample cohort.** Fresh frozen tumor samples (PTs and IBTRs) from 385 patients operated with BCS with and without radiotherapy in three health care regions

(Southern Sweden, Uppsala-Örebro, and Stockholm) were previously collected in a multi-center cohort, previously analyzed by gene expression[24]. We selected samples based on availability for downstream DNA, RNA, and protein extraction, routine biomarkers (Estrogen Receptor, ER; Progesterone Receptor, PgR, Heregulin 2, Her2/ERBB2; proliferation marker Ki-67), follow-up information until formation of IBTR (IBTR-free survival, IBTRFS), and availability of formalin-fixed and paraffin-embedded (FFPE) material for re-analysis. A total of 54 samples (27 PTs matched by 27 IBTRs) was selected (Supplementary Data 1). Samples for germline DNA whole genome sequencing (WGS) were available for two patients (S12 and S18). All surgically removed PTs were evaluated for residual disease and the margins were reported free from invasive cancer or ductal carcinoma in situ with a minimum of 10 mm healthy tissue around the tumor, according to Swedish national guidelines at the time of surgery.

An additional cohort of 21 primary tumors from patients who did not develop any recurrent disease were also included in this study and analyzed by RNAseq and MS. Clinical marker stainings were performed anew for these samples (Supplementary Data 2). All specimens to be used for DNA, RNA, and protein extraction were stored as fresh frozen samples. All specimens analyzed in this project are under approval from the Ethical Review Board (Etikprövningsnämnden) with numbers DNR LU-2010/127 and LU-2001/240. Oral and written information was given to the patients and oral consent was obtained according to the study protocol.

**Immunohistochemical routine biomarker analysis.** FFPE tissues were cut into 3–4 µm sections and put on TOMO slides (MG-TOM-11/90, Histolab). Evaluation of ER (immunohistochemistry, IHC; staining cutoff: 10% positive cells), PgR (IHC; staining cutoff: 10% positive cells), ERBB2/Her2 (IHC and in situ hybridization for equivocal cases) and Ki-67 (staining cutoff: 30% positive cells) were performed according to routine clinical practice in Sweden. Briefly, antibodies used for IHC stainings were: ER: clone SP1, 790-4324 Ventana, concentration 1 µg/mL; PgR: clone 1E2, 790-2223 Ventana, concentration 1 µg/mL; Ki-67: clone MIB-1, M7240 DAKO (dilution 1:100), concentration 46 µg/mL; HER2: clone 4B5, 790-2991, concentration 6 µg/mL. Slides for ER, PgR, and Ki-67 evaluation were stained on the Discovery ULTRA (Ventana Medical System Inc., Tucson, AZ, USA). Her2 staining was performed on the Benchmark ULTRA (Ventana Medical System Inc., Tucson, AZ, USA). For all IHC, ULTRA cell conditioning (ULTRA CC1) pH 8–8.5, was used for heat-induced epitope retrieval. The primary antibodies were incubated for 32 min and visualized with conventional 3,3′-diaminobenzidine IHC detection kit. Leukocyte counts were evaluated on hematoxylin-eosin stained tissues by an experienced pathologist (GP). The histo-score combining the percentage of Ki-67 stained cells and the percentage of infiltrating leukocytes was calculated as follows:

$$Histoscore = \frac{Ki67 - mean(Ki67)}{sd(Ki67)} + \frac{Leukocytes - mean(Leukocytes)}{sd(Leukocytes)} \quad (1)$$

**DNA, RNA, and protein extraction.** Breast tumor tissues (PTs and IBTRs) were processed using the AllPrep DNA/RNA/Protein (Qiagen) protocol. Tissue lysis was performed by re-suspending ~30 mg of sliced frozen tissue in a solution containing 1% β-mercaptoethanol in RLT buffer (supplemented with antifoam agent; ID 19088, Qiagen). Next, steel beads (ID 79656, Qiagen) were added and samples were incubated in a Tissue Lyser LT (Qiagen) for 4 min at 50 Hz. Steel beads were then removed and 400 µL of 1% β-mercaptoethanol in RLT buffer was added to samples, which were then centrifuged at 14,000 × g for 5 min. Supernatants were transferred to new tubes, and then frozen at −80 °C. DNA, RNA, and protein extraction were performed according to manufacturer instructions (AllPrep DNA/RNA/Protein minikit; Qiagen). Each spin column flowthrough (DNA, RNA, protein) was stored at −80 °C until analysis (sequencing or mass spectrometry; MS).

**Whole-genome sequencing.** Sample library was performed twice for every sample, using a PCR-free method for specimens with high DNA yield, and employing a PCR amplification step for low yield samples. PCR-free libraries were prepared from 1 µg DNA using the TruSeq PCRfree DNA sample preparation kit (cat# FC-121-3001/3002, Illumina) targeting an insert size of 350 bp. PCR-amplified sequencing libraries were prepared from 100 ng DNA using the TruSeq Nano DNA sample preparation kit (cat# FC-121-4001/4002, Illumina) targeting an insert size of 350 bp. Both library preparations were performed according to manufacturers' instructions. Paired-end DNA sequencing with 150 bp read length was performed at the SNP&SEQ Technology Platform in Uppsala (Uppsala University, Uppsala, Sweden) using an Illumina HiSeqX sequencer (Illumina, San Diego, CA) with v2.5 sequencing chemistry.

**Variant calling.** Alignment to reference genome GRCh38 was performed using bwa's (v0.7.13) BWA-MEM algorithm, and conversion to BAM format and coordinate sorting was performed using samtools (v1.3). Duplicates were marked using Picard (v2.0.1). To identify all possible active regions and ensure that all samples had their information represented comparably the tools RealignerTargetCreator and IndelRealigner from GATK (v3.7) were used. Samples were processed with a scatter-gather methodology, dividing each sample by chromosome to

identify and realign any misaligned reads in active regions. Samples were then merged using Picard MergeSamFiles. GATK 3.7 BaseRecalibrator and PrintReads were used to identify potential systematic errors in the data and recalibrate the base quality scores.

A panel of normal (PoN) variants was created and used as a blacklist during variant calling. First, variants were called using GATK (v3.8) MuTect2 with only a normal sample as input and then CombineVariants to aggregate the output for all normal samples. Only variants observed in at least two samples were included. For further variant calling, pairs of matched normal and tumor samples (2 out of 27 patients: S12 and S18) were called together if the patient had a matched normal sample, otherwise the tumor sample was called alone. In either case the PoN was used. A scatter-gather methodology was used to optimize runtimes, and CatVariants was used to merge the variants. The variants were filtered using the built in Mutect2 filtering (cutoff: lack of PASS annotation).

The variants were annotated using snpEff[59] (v4.2) and annovar (v2017.07.16). Information on the allele frequency of variants in population databases SweGen, ExAC and gnomAD was also added together with COSMIC database annotation. For the SweGen and ExAC database annotation, a lift-over of the variant files was performed using Picard (v2.10.3) with the LiftoverVcf command to obtain GRCh38 coordinates.

We applied the TPES[34] (v1.0.0; https://cran.r-project.org/web/packages/TPES/index.html) package to estimate tumor purity values for all cancer samples using the single nucleotide variant (SNV) list as input. These estimates were used to filter the SNVs.

The filters are applied in order as follows:

Population allelic frequency filter: the observed allelic frequency in gnomAD and SweFreq needed to be 0 or NA (i.e., variant has never been observed in a sample from these datasets).

Allelic Depth (Support) filter: the number of reads supporting the variant in the tumor needed to be larger or equal to 2. Additionally, if a matched normal was available, the number of reads supporting the variant in that sample needed to be 1 or 0.

Coverage filter: the total number of reads overlapping the position needed to be 10 or more in the tumor sample. If a matched normal sample was available, that needed to have 10 or more reads coverage.

PoN filter: variant not present in the PoN file (i.e., it cannot have been detected in any of the normal samples from this dataset).

TPES filter: the Log2 ratio of the probability of a given variant being observed under the cancer vs. background model needed to be ≤ −1 (i.e., removal of variants where the background model is twice as likely to produce the observed variant). Specifics on calculations are presented here:

For each sample $i$ with a TPES[34] based tumor purity estimate ($t_i$) we define

$$p_i^{cancer} = 0.5 * t_i \quad (2)$$

$$p_i^{background} = 0.5 \quad (3)$$

as the probabilities of a binomial distribution for the heterozygous SNV model, analogous with 1.0 instead of 0.5 for the homozygous SNV model. If no TPES based tumor purity estimate ($t_i$) existed for a given sample $i$, this filtering step was skipped.

We then calculate for each variant $j$ the ratio

$$Ratio = 2\frac{P\left(s_j; c_j, p_i^{cancer}\right)}{P\left(s_j; c_j, p_i^{background}\right)} \quad (4)$$

where $s_j$ is the support of variant $j$ in sample $i$ and $c_j$ is the coverage of that variant. Here, $P(k;n;p)$ is the probability of observing exactly $k$ out of $n$ hits (i.e. reads with alternative allele) under a binomial distribution with probability $p$.

If the Log2 ratio is larger (>) than −1 for the heterozygous or the homozygous case, the variant is kept. If it is below or at −1 in (≤) both in the homo- and heterozygous cases, indicating a higher probability in the background model, the variant is filtered out.

The SNV list was derived by extracting the mutations contained in the union of the COSMIC cancer gene census[60], the FoundationOne® gene list[61], genes part of the Memorial Sloan Kettering IMPACT platform[62], and the list of reported BC driver genes[12]. Variants were filtered based on impact (moderate or high were included) and type of variant (downstream gene variant, upstream gene variant, 3′ UTR variant, 5′ UTR variant, and synonymous variant cases were excluded).

Mutational signatures were determined using the MutationalPatterns package (v3.3.0)[63] by fitting the SNV counts per 96 tri-nucleotide context to the 30 COSMIC signatures (v2; https://cancer.sanger.ac.uk/signatures/signatures_v2/)[60]. Mutational signature contributions were reported as fractions. The resulting table was then filtered for signatures with enough contribution (signatures within the 1st and 2nd quartiles were selected based on mean contribution across samples).

We used sciClone (v1.1.0; https://github.com/genome/sciclone) to build the clustering of SNVs by their variant allele frequencies. Applying clonevol (v0.99.11; https://github.com/hdng/clonevol) to sciClone-derived clustering did not yield any valid model of tumor evolution.

**Copy number call**. CN calls were obtained by determining total coverage across the genome in 10 kb bins for each sample, then using the R locfit.robust function to fit the relationship between GC content and bin coverage, then adjusting for the differences in GC-coverage relationships across samples.

The resulting adjusted coverage values were converted into Log2 ratios by employing either the matched normal sample (if available), or the median adjusted coverage of the sample itself as denominator. The Log2 ratios were then centered (median subtraction), adjusting for between-sample coverage differences.

CN segmentation was performed on the centered Log2 ratios using the circular binary segmentation algorithm implemented in the DNAcopy R/Bioconductor package. The resulting CN segments were then mapped to genes by finding overlaps with annotated exons of each gene. For genes overlapping multiple copy number segments the CN values were averaged.

To determine CN gains and losses between paired PTs and IBTRs a CN delta was calculated with the following formula:

$$DeltaCN = CN(IBTR) - CN(PT) \qquad (5)$$

CN changes were taken into account only if they impacted genes that showed a minimum CN of 0.5 in matched PT and IBTR samples and where DeltaCN was above 0.75 (gain) or below −0.75 (loss).

**RNA sequencing**. RNAseq was performed as previously reported[19]. Briefly, the amount, concentration and quality of the extracted RNA was tested using a Bioanalyzer 2100 instrument (Agilent Technologies), a NanoDrop ND-1000 spectrophotometer (Thermo Fisher Scientific) or Caliper HT RNA LabChip (Perkin Elmer). All samples had a RNA integrity value of 6.0 or higher.

RNAseq library preparation and analysis were conducted as previously described[64]. Briefly, 100 ng of RNA input was used for cDNA library preparation using the TruSeq® Stranded mRNA NeoPrep kit (Illumina), according to manufacturer instructions. Concentration of cDNA was measured (QuantIT® dsDNA HS Assay Kit; Thermo-Fisher), and libraries were then denatured and diluted according to the NextSeq® 500 System Guide (Illumina). RNAseq was then performed on a NextSeq 500 (Illumina) sequencer generating paired-end reads of length 75 bp.

**RNAseq data processing**. De-multiplexed RNA-Seq reads were aligned to the GRCh38 human reference genome using STAR aligner (v020201) with an over-hang value of 75 to match the read-length. The standard GATK analysis pipeline was then applied (GATK; v3.7-0-gcfedb67). The resulting alignment files were processed by first generating per-gene read counts mapping to the GRCh38 GTF file from Ensembl (v95) using the *summarizeOverlaps* function in "Union" mode to count reads that uniquely mapping to exactly one exon of a gene (GenomicAligner, v1.18.1). Next, genes with no counts in any of the samples were discarded.

**Protein digestion**. Protein flow-throughs from the AllPrep protocol were precipitated in ice-cold (−20 °C) methanol, as previously described[65]. Briefly, protein pellets were then suspended in 100 mM Tris (pH 8.0) buffer containing 100 mM dithiothreitol and 4% w/V sodium-dodecyl-sulphate and incubated at 95 °C for 30 min under mild agitation. Samples were then cooled to room temperature, diluted in 8 M urea in 100 mM Tris (pH 8.0) buffer, loaded on 30 KDa molecular filters (Millipore) and centrifuged at 14,000 × g for 20 min. Filters were washed with urea buffer and centrifuged at 14,000 × g for 10 min. Proteins were alkylated with iodoacetamide in urea buffer (30 min in the dark), washed with urea buffer and tri-ethyl-ammonium bicarbonate buffer (pH 8.0), and trypsin was added (enzyme-protein ratio 1:50; incubation at 37 °C for 16 h, 600RPM). Filters were then centrifuged at 14,000 × g for 20 min to retrieve tryptic peptides, loaded onto C18 (3 stacked layers; 66883-U, Sigma) stage tips (pretreated with methanol, 0.1% formic acid (FA) in 80% acetonitrile solution, and 0.1% FA in ultrapure water), washed with 0.1% FA in ultrapure water solution, and eluted with 0.1% FA in 80% acetonitrile. Eluates were then dried and subjected to SP3 peptide purification, as previously described[66]. Briefly, 2 μL of SP3 beads (1:1 ratio of Sera Mag A and Sera Mag B re-suspended in ultrapure water; Sigma) were added to dried peptides and incubated for 2 min under gentle agitation. A volume of 200 μL of acetonitrile was then added and samples were incubated for 10 min under agitation. Sample vials were then placed on a magnetic rack and washed again with acetonitrile for 10 min. Elution was performed by adding 200 μL of 2% dimethyl sulfoxide in ultrapure water to the bead-peptide mixtures and incubating them for 5 min under agitation. Supernatants were then collected, dried, and stored at −80 °C until MS analysis.

**Mass spectrometry analysis**. Tryptic peptide mixtures were subjected to data-independent acquisition MS analysis. Samples were eluted in a 120 min gradient (flow: 300 nl/min; mobile phase A: 0.1% FA in ultrapure water; mobile phase B: 80% acetonitrile and 0.1% FA) on a Q-Exactive HFX (Thermo-Fisher) instrument coupled online to an EASY-nLC 1200 system (Thermo-Fisher). Digested peptides were separated by reverse phase HPLC (ID 75 μm × 50 cm C18 2 μm 100 Å resin; Thermo-Fisher). Gradient was run as follows: 10–30% B in 90 min; 30–45% B in 20 min; 45–90% B in 30 s, and 90% B for 9 min. One high resolution MS scan (resolution: 60,000 at 200 m/z) was performed and followed by a set of 32 data independent acquisition MS cycles with variable isolation windows (resolution:

30,000 at 200 m/z; isolation windows: 13, 14, 15, 16, 17, 18, 20, 22, 23, 25, 29, 37, 45, 51, 66, 132 m/z; overlap between windows: 0.5 m/z). Ions within each window were fragmented by HCD (collision energy: 30). Automatic gain control target was set to 1e6 for both MS and MS/MS scans, with ion accumulation time set to 100 ms and 120 ms for MS and MS/MS, respectively. Protein intensities were derived by employing our previously established computational workflow[19]. A total of 4,640 proteins were identified after FDR filtering (cutoff: 0.01). Batch effect correction was performed using the limma[67] (v3.46.0) package. Raw protein intensities were Log2 transformed and centered prior differential expression analysis.

**Statistics and reproducibility**. All statistical tests were performed in R (v4.0.5; correlations, hierarchical clustering, and differential expression tests) or Graph-PAD (v9; contingency tables for Fisher and Chi-square tests). Total sample size of the dataset was 75 (27 PTs; 27 IBTRs; 27 PTs with no recurrence).

Differences in mutational signatures contribution across clinical variable-grouped samples were assessed by Wilcoxon Mann-Whitney rank sum test, with the resulting p-values adjusted for multiple comparison using the Benjamini-Hochberg method.

Matched PT-IBTR dissimilarity coefficients were calculated on complete feature for CN (n features: 59,100), RNA (n features: 12,750), and protein (n features: 4,640) tables using cosine similarity using the lsa package (v0.73.3) to correct for differences based on different feature numbers. Dissimilarities were then calculated with the following formula:

$$Dissimilarity = 1 - Cosine\ similarity \qquad (6)$$

Gene Set Enrichment Analysis (GSEA; v4.1.0)[68] was performed on scaled and Log2 transformed RNA and protein tables. Databases: Hallmarks (v5.2), ALL (v5.2; KEGG subset was then selected); permutation type: gene set; scoring: classic; metric: t test; other parameters were kept at default settings; significance cutoff: FDR < 0.25. GSVA scores were calculated for selected Hallmarks gene sets (i.e., E2F targets and allograft rejection) derived from GSEABase (v1.52.1) using the GSVA package (v1.38.2)[69]. Survival analyses were performed using the survival package (v3.2.10) and Kaplan–Meier curves were plotted with the survminer (v0.4.9) package. Classes cutoffs for survival analyses were determined using median expression (GSVA score or histo-score). For all statistical tests, p values below (<) 0.05 were considered significant. All statistical tests were two-sided.

**Reporting summary**. Further information on research design is available in the Nature Portfolio Reporting Summary linked to this article.

## Data availability

The DNA and RNA sequencing data are not publicly available due to ethical considerations with regards to person-identifying information (GDPR) and due to prohibitions by Swedish law. The raw counts from the RNAseq were uploaded on FigShare together with data referring to all included figures (https://doi.org/10.17044/scilifelab.c.6387480.v1). Source data underlying figures is also provided in Supplementary Data 4. Data-independent acquisition MS data, and their respective search result files have been deposited to the ProteomeXchange Consortium via the PRIDE partner repository[70] with the dataset identifiers PXD032266 (matched PT-IBTR set) and PXD037428 (PTnorec set).

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

## Acknowledgements

We gratefully thank Sara Baker, Carina Forsare, Kristina Lövgren and Anna-Lena Borg for excellent technical assistance. We also thank the biobanks of the South Sweden Breast Cancer Group, the Biobank at the Department of Oncology and Pathology Lund University biobank at Cancer Center Karolinska and the Biobank at Academic Hospital in Uppsala and Department of Pathology, Uppsala University, for collecting the samples and making them available for studies. Sequencing was performed by the SNP&SEQ Technology Platform in Uppsala. The facility is part of the National Genomics Infrastructure Sweden and Science for Life Laboratory. The SNP&SEQ Platform is also supported by the Swedish Research Council and the Knut and Alice Wallenberg Foundation. Parts of the computational analysis was performed on resources provided by SNIC through Uppsala Multidisciplinary Center for Advanced Computational Science under Project sens2016010 and the authors would like to acknowledge support from Science for Life Laboratory, the National Genomics Infrastructure, National Bioinformatics Infrastructure Sweden and Uppsala Multidisciplinary Center for Advanced Computational Science for providing assistance in massive parallel sequencing and computational infrastructure.

The study was made possible with support from the Marianne and Marcus Wallenberg Foundation, the Swedish Breast Cancer Association (BRO), the Swedish Cancer Society (Cancerfonden), Region Skåne, Governmental Funding of Research within the Swedish National Health Service (ALF), Mrs. Berta Kamprad Foundation, Anna-Lisa and Sven-Erik Lundgren Foundation, Magnus Bergvall Foundation, the Gunnar Nilsson Cancer Foundation, BioCARE, the King Gustaf V Jubilee Fund, and Bergqvist Foundation.

## Author contributions

T.D.M., P.P., M.S., J.M., and E.N. designed the study. T.D.M., M.S., P.P., S.R., B.N., and S.L. processed sequencing data. T.D.M. and P.P. processed proteomic data. L.T. and G.P. performed and scored immunohistochemical stainings. M.S., F.W., I.F., P.M., L.M., J.M., M.F., and E.N. administered medical records, provided analysis platforms and support. T.D.M. wrote the manuscript with assistance from all authors.

## Funding

## Competing interests

P.M. and M.F. research contract with PFS Genomics. Remaining authors have no competing interest.
