## [Peer Review File · Communications Biology]

Reviewers' comments:

Reviewer #1 (Remarks to the Author):

In the manuscript "Evolution of ipsilateral breast cancer decoded by proteogenomics" submitted by De Marchi et al. the authors performed a proteogenomics analysis of 27 matching primary breast cancers and ipsilateral breast tumor recurrence (IBTR). The goal was to identify if IBTRs are associated with molecular changes that could be explored for precision medicine. The analysis revealed a relationship between estrogen and progesterone receptors and increased levels of genetic changes. The authors conclude that enhancing the genomic instability in some tumors could serve as an alternative treatment option. The manuscript is well written and present a large amount of data (total of 17 Figures) on a relevant clinical/biological topic.

Major issues:

- "We observed that distances between PT-IBTR pairs in general were greater at the RNA and protein levels when compared to CN and SNV (Figure S12A)". This is expected since the authors used Euclidean distance, which depends on the number of features (dimension) and numeric range of the features. This does not necessarily indicate larger repercussion at expression level. If the authors want to show this, they have to use a measure which is not dependent on the number of features and numeric range.
- Most of the discussed effect sizes (e.g. Fig 2D, Fig 6A/B) are very small and would normally not survive filtering (e.g. Fig 6A/B no filter for fold change). For example, the reported log₂ fold changes in DNA repair between PT and IBTR is something in the range of 0.25, which well within the variation one would expect for technical replicates. Please discuss the relevance of the observed changes.
- In my view the authors have overloaded the manuscript and supplementary information with figures which are not helping to build up a consistent story. I strongly recommend the authors to reconsider the choice and number of plots.

Minor issues:

- Fig 2B: Y-axis describes that the top 5 signatures are shown but 10 are visible. Signature 17 (?) shows no change (median ~0, variance ~0)? Why is this among the top 5 by median contribution change?
- Fig 2D: Differing number of samples used for testing. Indicate n. Justify why. Indicate in y axis that this is only Signature 3
- Missing "." before sentence "Upon assessing the most frequently mutated genes within ER positive and negative tumors ..."
- Fig 6: I saddest to switch the PT/IBTR and the RNA/Protein label to allow easier comparison between the relevant classes. Right now, the focus is on the change between RNA to protein within e.g. PT (not relevant here). Also, why are the log₂ fold changes at most -1/1? Is this an artefact? Also, the effect size appears very small. Discuss the relevance of this.
- Fig S10: The visualization of the distance matrix for the SNV data looks odd. Why does the Euclidean distance stop at 2 and why is there next to no variability across the samples?
- Fig S10: The visualization could be improved by using "NA" for identical samples (diagonal). This will increase the effective range of the color key and will show differences more clearly.

Reviewer #2 (Remarks to the Author):

The authors performed whole genome sequencing, RNA sequencing, and a mass spectrometry-based proteome analysis utilizing a total of 54 breast tumors consisting of 27 primary and their paired recurrent tumors. These tumors recurred following surgical resection of the primary tumor plus adjuvant radiotherapy (70%), endocrine therapy (30%), or chemotherapy (22%). It was the aim of their study to investigate the evolution of ipsilateral breast cancer. Most of the reported observations are based on mutational signature analysis of unpaired tumors (adjacent non-cancerous tissue or another source of germline DNA was missing) which is a suboptimal approach. The presented proteomics data are mostly from routine IHC data, and the overall presentation of the findings is

rather convoluted. Most importantly - it is quite difficult to catch a key message from the findings and this reviewer has a hard time to capture what is learned from the data that would help us in identifying primary breast tumors with an increased risk of developing the ipsilateral recurrent disease. Lastly, the methods and figures are poorly described, and we do not learn how many of the analyses were done.

Comments

This reviewer is not convinced there is "an evolution" of ipsilateral recurrent disease. Instead, there is residual disease after resection that is treated and becomes recurrent, perhaps because of certain tumor characteristics that are already present in the primary tumor. If there is robust evolution, this manuscript does not provide the evidence.

It is a key question why some breast tumors relapse, and why others not. Is there a specific biology? This reviewer would expect that one would compare the biology of primary breast tumors from patients with ipsilateral recurrence versus primary breast tumors from patients who did not experience a disease recurrence. Is there a difference in tumor biology and/or the mutational signature between those tumors that explains why some breast tumors have an increased risk of recurrence? This is not what was done in this manuscript – a weakness.

Nakagomi et al. (BCRT 2022: 193, 349-359) studied mutational signatures in ipsilateral breast cancer recurrence and used an interesting approach comparing recurrent disease with double primaries, meaning a patient developed a primary breast tumor twice in their lifetime. They found that recurrent breast tumors were significantly more likely to harbor a PIK3CA-Akt pathway mutation than a double primary tumor and propose that these mutations could be key driver for disease recurrence. Question: What is the PIK3CA-Akt pathway mutation frequency in the Swedish cohort of primary and recurrent tumors and how does it compare to the frequency in breast tumors from patients without recurrence (the data could be obtained from TCGA)?

Abstract: What information presented in the abstract is specific to the research question why ipsilateral recurrent breast cancer may develop or provides evidence of a possible underlying mechanism for ipsilateral recurrent breast cancer? Hard to catch. The current statements are rather general and would broadly apply to breast cancer.

What is the definition of a recurrent tumor versus double primary for this study?

Did the authors apply paired analysis statistical tests, as they should?

Figure 1: What is shown in 1E? What do these data mean?

The authors describe a significant difference in Ki-67 levels between primary and the paired recurrent tumors. Table 1 does not show this difference. Perhaps authors use the gene expression data and a proliferation gene expression signature to show that the proliferation index (z-score-based) of these tumors is indeed different. Those signatures can be found in the published literature and provide continuous data for the proliferation index of a tumor.

This reviewer does not find that much is learned from the mutational signature analysis. It describes an etiology by association. In addition, the mutational signatures are difficult to interpret as paired germline DNA was not analyzed for most of the tumors – a weakness. It would be more important to identify driver mutations that underly the development of recurrent disease which is missing.

Mutational signature, as described, may well develop because of the applied cancer therapies but may not at all be drivers of disease progression. Also, the focus on relationships between mutational signatures and tumor marker expression, as done in the manuscript, appears unfocused and does not help in explaining why these ipsilateral recurrent tumors developed.

Figure 2 and the associated text under results. How was the mutational signature analysis performed, how was the statistical analysis performed, how did the authors adjust in the analysis, how was the analysis corrected for multiple comparison analysis? There is nothing in the methods. More generally, the reviewer has a hard time to understand what is shown in the different graphs that make up Figure 2. Just all confusing! Where is the contrast primary versus recurrent disease in Figure 2?

Figure 3: Again - the reviewer cannot understand what is shown in this figure. Lots of colors but no explanations. Which of the shown data relates to the contrast primary versus recurrent disease?

Changes in copy number and key drivers related to what? What are the key drivers?

Tumor evolution: How were these analyses performed? Not described under methods.

Figure 4; some of the analysis shown in this figure should be the pre-requisite of the study, assuring

that the recurrent tumors are indeed recurrent disease and not double primaries – a completely different disease with a very different prognosis. Once again, how do the shown data help us understanding why the primary tumors gave rise to recurrent disease – the key question? At different places in the manuscript write-up the tumor ER status has something to do with the evolution of ipsilateral recurrent breast tumors. Just to be sure: ER-negative and ER-positive breast cancer are quite different diseases beyond the tumor ER status. It should be known from the literature if ER-negative breast cancer is more likely to develop ipsilateral recurrent tumors. Figure 5, again, how do the shown data help us understanding the risk of developing ipsilateral recurrent tumors, other than there is something different in our data by ER status. Its all confusing, missing a cohesive approach to understand the transition from primary to recurrent disease and its key drivers. Same applies to Figure 6.

Lastly, the authors mention on page 26 that the usage of specimen for research is under approval. This must be a mistake. This research should not have taken place if not approved.

Reviewer #1 (Remarks to the Author):

In the manuscript “Evolution of ipsilateral breast cancer decoded by proteogenomics” submitted by De Marchi et al. the authors performed a proteogenomics analysis of 27 matching primary breast cancers and ipsilateral breast tumor recurrence (IBTR). The goal was to identify if IBTRs are associated with molecular changes that could be explored for precision medicine. The analysis revealed a relationship between estrogen and progesterone receptors and increased levels of genetic changes. The authors conclude that enhancing the genomic instability in some tumors could serve as an alternative treatment option. The manuscript is well written and present a large amount of data (total of 17 Figures) on a relevant clinical/biological topic.

Major issues:

- “We observed that distances between PT-IBTR pairs in general were greater at the RNA and protein levels when compared to CN and SNV (Figure S12A)”. This is expected since the authors used Euclidean distance, which depends on the number of features (dimension) and numeric range of the features. This does not necessarily indicate larger repercussion at expression level. If the authors want to show this, they have to use a measure which is not dependent on the number of features and numeric range.

We thank the reviewer for this comment and we agree that Euclidean distances were a sub-optimal choice to display pairwise shifts between matched primary and recurrent tumors. In the new version of the manuscript we used the Cosine dissimilarity (i.e. 1-Cosine similarity) and have updated Figures 4, S6, and S8 and the related text in several places (Methods: p 35, lines 677-81; Results: p 17, lines 284-97; p 18, lines 305-20; Discussion p 22, lines 406-10).

- Most of the discussed effect sizes (e.g. Fig 2D, Fig 6A/B) are very small and would normally not survive filtering (e.g. Fig 6A/B no filter for fold change). For example, the reported log 2 fold changes in DNA repair between PT and IBTR is something in the range of 0.25, which well within the variation one would expect for technical replicates. Please discuss the relevance of the observed changes.

We agree with the reviewer on this. The small effect sizes that are shown in Figure 2 pertain to changes in molecular signature/s contributions, which in turn were derived from variant calling. Sample collection for this study was initiated in 1990s with the idea to obtain the longest possible follow up and samples of recurring tumors on top of primary ones. Collection of normal tissue or blood draws were not included in the standard clinical practice of the time. This limited availability of specimens resulted in several tumor samples lacking a matching normal adjacent tissue or blood sample, which impacted the ability to filter germline polymorphisms. This meant that the signal reported as mutational signature contribution was likely impacted by this, hence the small effect. In the original plots in Figure 5 and 6 (we reported transcript and protein fold changes for both significant and non-significant transcripts/proteins, explaining why the log 2 fold changes was

rather small. As the reviewers urged us focus on significant changes, we have removed panels A and B from Figure 5 in the modified version of the manuscript and focused on the pathway panels. We have clarified this in the Figure legend and added the aforementioned points at the end of the discussion section as limitations of this study (p 24, lines 449-54).

- In my view the authors have overloaded the manuscript and supplementary information with figures which are not helping to build up a consistent story. I strongly recommend the authors to reconsider the choice and number of plots.

We agree with this comment, thus we have reduced the amount of Supplementary Figures.

Minor issues:

- Fig 2B: Y-axis describes that the top 5 signatures are shown but 10 are visible. Signature 17 (?) shows no change (median ~0, variance ~0)? Why is this among the top 5 by median contribution change?

We apologize for the mistake and we have corrected Figures 2B and 2C. Top 5 signatures were selected based on absolute value mean contribution delta across samples. As the contribution for signature 17 was only shown in a few samples, we removed this from association analyses. We have clarified this in the text (p 14; lines 228-30).

- Fig 2D: Differing number of samples used for testing. Indicate n. Justify why. Indicate in y axis that this is only Signature 3

The different numbers of samples used in testing for association to clinical variables is because we have missing values for some of the clinical features e.g. PgR status evaluation by immunohistochemistry. We have amended Figure 2 and extended its legend. All other Figures that included similar analyses were also modified accordingly.

- Missing “.” before sentence “Upon assessing the most frequently mutated genes within ER positive and negative tumors ...”

We have amended this.

- Fig 6: I saddest to switch the PT/IBTR and the RNA/Protein label to allow easier comparison between the relevant classes. Right now, the focus is on the change between RNA to protein within e.g. PT (not relevant here). Also, why are the log2 fold changes at most -1/1? Is this an artefact? Also, the effect size appears very small. Discuss the relevance of this.

We have switched the facets of Figure 6A and 6B (now in Figure 5) to better display the changes between PTs and IBTRs. As stated in the response above, we would like to clarify that the genes-proteins reported in these panels are comprised of both significant and non-significant transcript/proteins explaining the small effect observed in these plots. We have added this information to the legend of Figure 5.

- Fig S10: The visualization of the distance matrix for the SNV data looks odd. Why does the Euclidean distance stop at 2 and why is there next to no variability across the samples?

We have changed Figure S10 (now Figure S6) in conjunction to switching to Cosine-based dissimilarity (i.e. 1-Cosine similarity) coefficients. Regardless of this, we believe that the very short distance between samples at the mutational level refers to the number of shared/unique mutations, which displays a significantly smaller number of features when compared to CN, transcript, or protein levels. Given the fact that mutation calling was impacted by the limited availability of normal tissue, thus prompting us to investigate mutations only in known cancer drivers, we have removed this analysis.

- Fig S10: The visualization could be improved by using “NA” for identical samples (diagonal). This will increase the effective range of the color key and will show differences more clearly.

We have re-plotted the heatmaps using Cosine dissimilarity coefficients, yet we believe that keeping the values showing complete similarity are helpful to appreciate the actual range of the differences.

Reviewer #2 (Remarks to the Author):

The authors performed whole genome sequencing, RNA sequencing, and a mass spectrometry-based proteome analysis utilizing a total of 54 breast tumors consisting of 27 primary and their paired recurrent tumors. These tumors recurred following surgical resection of the primary tumor plus adjuvant radiotherapy (70%), endocrine therapy (30%), or chemotherapy (22%). It was the aim of their study to investigate the evolution of ipsilateral breast cancer. Most of the reported observations are based on mutational signature analysis of unpaired tumors (adjacent non-cancerous tissue or another source of germline DNA was missing) which is a suboptimal approach.

We apologize if we were not clear in the text. We understand that lack of normal tissues constitutes a limitation for mutation calling, which is why we focused on the genomic changes between matched primary and recurrent tumors. We have specified this in the text, as well as expanding the Discussion section on the limitations of our study (p 24, lines 449-54).

The presented proteomics data are mostly from routine IHC data, and the overall presentation of the findings is rather convoluted.

We have amended the text (Results and Discussion) so to clarify the main findings and prune superfluous or redundant text.

Most importantly - it is quite difficult to catch a key message from the findings and this reviewer has a hard time to capture what is learned from the data that would help us in identifying primary breast tumors with an increased risk of developing the ipsilateral recurrent disease.

We addressed this point by analyzing a cohort of primary breast tumors from patients who did not develop any recurrence (local or otherwise) within 10 years from surgical resection. Clinical characteristics are reported in Table S2, while comparison between the two sets of primary tumors is shown in Table S3. This cohort comprised 21 patients analyzed by RNAseq and mass spectrometry. We have expanded the Introduction (p 12, lines 174-6 and 182-4) Methods (p 26, lines 475-7), Results (p 20-1, lines 363-87), and Discussion (p 24, lines 441-7; p 25, lines 458-60) sections in relation to the analysis of these primary tumors to the ones that developed an IBTR.

Lastly, the methods and figures are poorly described, and we do not learn how many of the analyses were done.

We have clarified the Methods section (p 26, lines 472-80; p 27, lines 494-7; p 31, lines 582-5; p 35-6, lines 674-90).

Comments

This reviewer is not convinced there is “an evolution” of ipsilateral recurrent disease. Instead, there is residual disease after resection that is treated and becomes recurrent, perhaps because of certain tumor characteristics that are already present in the primary tumor. If there is robust evolution, this manuscript does not provide the evidence.

We thank the reviewer for this comment as it points out confusing parts of our manuscript. Primary tumors and IBTRs were surgically resected with 10 mm tumor margins. In all tumors the margin was checked for presence of residual tumor cells or ductal carcinoma *in situ*. All of these margins were found negative for cancerous cells, thus excluding the possibility of residual disease. We have added this in our Methods section (p 26, lines 472-4).

On top of this, we did not use the term “tumor evolution” in relation to cancer cell subpopulation dynamics (clonal sweeps, expansions, etc.), but to describe how IBTRs genomes, transcriptomes, and proteomes differed from their matched primary tumors. We understand now our wording might generate confusion so we have changed this throughout the text.

It is a key question why some breast tumors relapse, and why others not. Is there a specific biology? This reviewer would expect that one would compare the biology of primary breast tumors from patients with ipsilateral recurrence versus primary breast tumors from patients who did not experience a disease recurrence. Is there a difference in tumor biology and/or the mutational signature between those tumors that explains why some breast tumors have an increased risk of recurrence? This is not what was done in this manuscript – a weakness.

We believe we have addressed this question by providing a comparative analysis of the aforementioned cohort of primary tumors that did not develop any recurrence. This analysis compared primary tumors that developed an IBTR (n = 27) with tumors from patients that did not experience any recurrence within 10 years after resection of the primary mass (n = 21). These samples were analyzed by RNAseq and proteomics and our results reveal proliferative and immune signaling networks to be highly enriched in primary tumors that develop IBTR, with the prospect of recapitulating our *omics* data using clinical evaluation methods, such as immunohistochemistry, for preventive screening.

Nakagomi et al. (BCRT 2022: 193, 349-359) studied mutational signatures in ipsilateral breast cancer recurrence and used an interesting approach comparing recurrent disease with double primaries, meaning a patient developed a primary breast tumor twice in their lifetime. They found that recurrent breast tumors were significantly more likely to harbor a PIK3CA-Akt pathway mutation than a double primary tumor and propose that these mutations could be key driver for disease recurrence. Question: What is the PIK3CA-Akt pathway mutation frequency in the Swedish cohort of primary and recurrent tumors and how does it compare to the frequency in breast tumors from patients without recurrence (the data could be obtained from TCGA)?

We have determined the frequency of PIK3CA and AKT mutations in our dataset (PTs and IBTRs) versus the TCGA one. In the TCGA dataset the mutational frequencies of PIK3CA and AKT were 426/1287 (33.10%) and 35/1287 (2.72%). In our dataset, PIK3CA displayed a mutational frequency of 4/27 (14.81%) in PTs and of 6/27 (22.22%) in IBTRs. We did not detect any AKT1 mutations. Fisher Exact test was employed to assess whether significant differences in PIK3CA mutation frequencies existed between the TCGA cohort and the Lund one (PTs and IBTRs). No significant difference was observed (TCGA vs Lund PTs Fisher $p = 0.060$; TCGA vs Lund IBTRs Fisher $p = 0.302$). This suggests that further investigation is needed to define which primary tumors are at risk of developing a local recurrence. Nonetheless, we have added a section in our Results (p 21, lines 376-80).

Abstract: What information presented in the abstract is specific to the research question why ipsilateral recurrent breast cancer may develop or provides evidence of a possible underlying mechanism for ipsilateral recurrent breast cancer? Hard to catch. The current statements are rather general and would broadly apply to breast cancer.

We have amended the abstract.

What is the definition of a recurrent tumor versus double primary for this study?

Currently there is no official definition of double primary and truly recurrent tumors. One way to help define this difference is to look at shared driver mutations, though that would require normal tissue for all specimens for unbiased variant calling. Based on this and what we wrote above, we assumed all IBTRs were true recurrences.

Did the authors apply paired analysis statistical tests, as they should?

The original analysis was not paired, but that only slightly affects the power of the analysis. Following Reviewer 1 comments, we removed PT-IBTR differential expression analyses and related panels.

Figure 1: What is shown in 1E? What do these data mean?

Figure 1 displays the verification of ER, PgR, Ki-67, and ERBB2 markers by “omics” measurements. ER, PgR and Ki-67 markers were evaluated by immunohistochemistry, while ERBB2 by a combination of immunohistochemistry and in situ hybridization. We then tested whether CN status (DNA WGS), transcript levels (RNAseq), and protein abundances (MS) matched immunohistochemistry results. For Figure 1E the CN (left), RNA (center), and protein (right) levels of ERBB2 are depicted. To explain this better, we have expanded the legend to provide more exhaustive information.

The authors describe a significant difference in Ki-67 levels between primary and the paired recurrent tumors. Table 1 does not show this difference. Perhaps authors use the gene expression data and a proliferation gene expression signature to show that the proliferation index (z-score-based) of these tumors is indeed different. Those signatures can be found in the published literature and provide continuous data for the proliferation index of a tumor.

We understand that the reviewer refers to Figure 1C and 1D, where transcript and protein levels of Ki-67 were tested between tumors with high and low expression as defined by immunohistochemistry. Conversely, Table 1 refers to differences in Ki-67 low/high expression (as defined by immunohistochemistry) frequencies between PTs and IBTRs. For this, the two analyses are not comparable. We understand that our text might be misleading and have amended it (p 13, lines 192-5).

This reviewer does not find that much is learned from the mutational signature analysis. It describes an etiology by association. In addition, the mutational signatures are difficult to interpret as paired germline DNA was not analyzed for most of the tumors – a weakness. It would be more important to identify driver mutations that underly the development of recurrent disease which is missing.

We agree with the reviewer that mutational signatures as they have been analyzed here, without matched normal DNA, are a suboptimal approach and that a key aspect in differentiating new primaries from actual recurrences is to define driver mutations. Unfortunately, lack of normal tissues hampered that effort as well, that is why we only focused on key cancer mutations derived from previous literature (e.g. Nik-Zainal et al). We apologize but this cannot be done.

Mutational signature, as described, may well develop because of the applied cancer therapies but may not at all be drivers of disease progression. Also, the focus on relationships between mutational signatures and tumor marker expression, as done in the manuscript, appears unfocused and does not help in explaining why these ipsilateral recurrent tumors developed.

We agree with the reviewer on the fact that mutational signatures might not be dependent on cancer therapies nor be drivers of cancer progression. Despite of this, some factors (clinical or otherwise) might be indicative of why some mutational signatures change in their contribution, hence the testing against clinical variables. We understand all this reporting might be confusing or even misleading, therefore we have opted to describe significant relationships only and have amended the related text (p 14, lines 219-25; p 14-5, lines 227-36), Figures, and Supplemental Figures.

Figure 2 and the associated text under results. How was the mutational signature analysis performed, how was the statistical analysis performed, how did the authors adjust in the

analysis, how was the analysis corrected for multiple comparison analysis? There is nothing in the methods.

We apologize for not clarifying this part in the Methods section. We have added the description of the tests we used (p 31, lines 582-5; p 35, lines 674-6).

More generally, the reviewer has a hard time to understand what is shown in the different graphs that make up Figure 2. Just all confusing! Where is the contrast primary versus recurrent disease in Figure 2?

Figure 2 represents the difference in mutational signature contribution (i.e. delta contribution) between paired samples. Still, we understand the Figure and the analyses depicted in it are confusing. First, given the fact that many signatures display little to no contribution across samples, we have repeated the analyses pruning signatures with no or minimal contribution across the dataset (p 31, lines 582-5; p 35, lines 674-6) so to assess only meaningful associations. Second, we have clarified the legends and amended the Figure so to display our results more clearly.

Figure 3: Again - the reviewer cannot understand what is shown in this figure. Lots of colors but no explanations. Which of the shown data relates to the contrast primary versus recurrent disease? Changes in copy number and key drivers related to what? What are the key drivers?

Figure 3 represents the change in CN and SNV (gain or loss) between matched PTs and IBTRs. Panels B and E (in the amended Figure) display significant association of CN and SNV gains with clinical tumor markers. We have amended the Figure legend.

Tumor evolution: How were these analyses performed? Not described under methods.

We have amended the text throughout the whole manuscript and clarified this.

Figure 4; some of the analysis shown in this figure should be the pre-requisite of the study, assuring that the recurrent tumors are indeed recurrent disease and not double primaries – a completely different disease with a very different prognosis. Once again, how do the shown data help us understanding why the primary tumors gave rise to recurrent disease – the key question?

Figure 4 referred to CN, mutational (SNV), transcriptomic, and proteomic changes between matched tumors, which has been collapse into dissimilarity vectors (now modified following Reviewer 1 comments). The results depicted in Figure 4 are indicative of how each IBTR drifted from their PTs in these dimensions and what factors might be responsible of that. To define which tumors are true IBTRs and not double primaries would require us to determine a predictor and to

validate it using genomic data using for example clonal analysis or definition of cancer driver mutations using matched normal tissue, which is unfortunately not possible. This is why we have not focused on this aspect but added the comparative analysis between recurring and non-recurring PTs, in which we have shown how immune infiltration and proliferative markers are associated to IBTR formation both at the *omics* (RNAseq and proteomics) and histological levels.

At different places in the manuscript write-up the tumor ER status has something to do with the evolution of ipsilateral recurrent breast tumors. Just to be sure: ER-negative and ER-positive breast cancer are quite different diseases beyond the tumor ER status. It should be known from the literature if ER-negative breast cancer is more likely to develop ipsilateral recurrent tumors.

We agree with the reviewer that ER positive and negative breast tumors display major differences at the biological and clinical level. To our knowledge, ER negative tumors have a greater tendency to establish recurrent disease, but not specifically IBTRs. When assessing the relationship between IBTR formation and ER status, Demicheli et al (Demicheli R et al, BMC Cancer, 2010) showed that the dynamics of IBTR formation are delayed in ER positive tumors, while Yi et al (Yi M et al, Ann Surg. 2011) and Purswani et al (Purswani JM et al, Wolrd J Clin Oncol, 2020) showed that ER status was not a risk factor for IBTR development. We have added these citations to our Discussion section (p 23, lines 416-9).

Figure 5, again, how do the shown data help us understanding the risk of developing ipsilateral recurrent tumors, other than there is something different in our data by ER status. Its all confusing, missing a cohesive approach to understand the transition from primary to recurrent disease and its key drivers. Same applies to Figure 6.

The analyses presented in Figure 5 and 6 (now merged) aimed at following up our previous observations on lack of ER being significantly associated to the accumulation of genomic changes in PT-IBTR pairs. Hence we hypothesized that these observations were followed by transcriptomic and proteomic reprogramming, which in turn showed enrichment of proliferation-related pathways in the ER negative PT-IBTR subgroup. Thus absence of ER expression was associated with a high frequency of genomic changes, the enrichment of specific mutational signatures, and high rates of proliferation that reverberated at the transcriptome and proteome level. Based on this we hypothesized that high proliferation rates could not be completely responsible for the differences we have observed at the genomic level, and that other mechanisms would affect the mutational load. Having observed an increase in DNA repair genes in ER negative IBTRs and no association of TP53 mutations with the frequencies of genomic changes, we assumed other factors were involved. That is why we focused on the APOBEC family, in particular on APOBEC3B. We clarified this in the text on p 20, lines 350-7).

RESPONSE TO REVIEWERS

Lastly, the authors mention on page 26 that the usage of specimen for research is under approval. This must be a mistake. This research should not have taken place if not approved.

We apologize for the confusion and have amended the text in the Methods section (p 26, lines 478-80).

Reviewers' comments:

Reviewer #1 (Remarks to the Author):

The authors have addressed my first major concern and partially the third, however, did not sufficiently address my second major concern.

Major issue:

- I was requesting a discussion about the relevance of the observed changes in e.g. now Figure 5 E and F. In my view, the figure shows that there is no difference between cell cycle and DNA repair expression between PT and IBTR neither on proteomics nor on transcriptomics level. The statement in the manuscript does not match the figure. The differences plotted appear insignificant. Why did the authors not calculate a p-value (q-value) for this? Why were "both significant and non-significant" transcripts and proteins picked for the analysis? At least in proteomics, a fold-change difference of a factor 2 is often not even considered to be biologically relevant.

- On this point but as a separate concern, is the relevance of the observed difference in signature contribution e.g. signature 3 and 9. The boxplots already show a substantial overlap and I am still very skeptical about the biological importance/relevance of this difference. The increase in signature 9 is (estimated by the figure) something around 0.1 on average. I have a hard time believing that this is relevant and not the result of random chance. I was trying to understand what the signatures actually contain and the reference used in the manuscript (from 2013) only lists 21 signatures, not the 30 investigated here. There are 4 signature types listed on COSMIC and a quick search did not allow me to figure out which set of signatures was used in this study. My reason for checking was to understand if e.g. signature 3 is already correlated to e.g. ER or PgR status. If signature 3 is in fact the same as the one postulated in the corresponding reference, prior work (<https://www.ncbi.nlm.nih.gov/pmc/articles/PMC2880433/>) already suggested that tumors with BRCA1/2 mutations are typically ER negative and PgR negative.

- With respect to signature 17: While the association analysis was removed, the fact that it remains (according to the figure, being the first listed there) the top signature with contribution changes although only 2 samples showed a difference, raises concerns about the ranking procedure and again the biological importance of the observed changes. This is further corroborated by the observation that a surprisingly large number of statistical tests performed in this study only result in "borderline" (terminology used in the manuscript when referring to C>T transitions with a p-value ~0.076) significance. This loops back to the signature 3, where a p-value of 0.074 testing the relation to ER expression is referred to as "significant". Examples for this are e.g. Fig S8 with 0 or 1 out of 9 tests and Fig 4 with 2 or 4 out of the 9 tests being significantly different when using a 0.01 or 0.05 cutoff, respectively.

Minor issues:

- Figure 2: There is a typo, referencing a panel D in the legend which does not exist (I assume it was meant to refer to new C). The legend still refers to the top 10 signatures in panel B although only 5 are shown.

- Figure 4: In the RNA clustering, a bit right of the center, two samples are labeled "matched" but belong to different groups. This must be a mistake in plotting/labeling matched pairs as the leaves of a dendrogram can be re-ordered without changing the clustering and thus the currently matched pair would not be plotted side-by-side anymore.

- The authors have addressed some of my concerns with respect to the complexity of the figures. However, I think this can still be further improved by e.g. not showing details which are not discussed in the manuscript. Most of the heatmaps contain annotations that are never discussed or even mentioned.

- The cosine dissimilarity measure used, particularly in Figure S6 C and B, suggests that the authors have applied the cosine similarity on scaled expression data, otherwise a value of 1.5 (essentially 1-(0.5)) is hard to explain. This may potentially explain the bi-modal distribution observed for RNA and protein expression data.

Reviewer #2 (Remarks to the Author):

The authors greatly improved the write-up of the manuscript. They comprehensively addressed the reviewers' comments, to the extent possible, and refocused on the research question asking why ipsilateral recurrent breast cancer may occur. They truly made an effort to improve the write-up of the manuscript.

Two minor comments:

Abstract, line 141, is genetic variants the right term or should it be somatic mutations?

Figure 2A, the heatmap looks like a white space - nothing in it. The authors may choose a better color coding to highlight the findings in this graph.

Reviewer #1 (Remarks to the Author):

The authors have addressed my first major concern and partially the third, however, did not sufficiently address my second major concern.

Major issue:

- I was requesting a discussion about the relevance of the observed changes in e.g. now Figure 5 E and F. In my view, the figure shows that there is no difference between cell cycle and DNA repair expression between PT and IBTR neither on proteomics nor on transcriptomics level. The statement in the manuscript does not match the figure. The differences plotted appear insignificant. Why did the authors not calculate a p-value (q-value) for this? Why were “both significant and non-significant” transcripts and proteins picked for the analysis? At least in proteomics, a fold-change difference of a factor 2 is often not even considered to be biologically relevant.

We thank the reviewer and we apologize for not addressing this properly. Panels E and F of Figure 5 were meant to show the change in distribution of Log2 fold change for cell cycle and DNA repair transcript/proteins between PT and IBTR samples. With regard to this we plotted the graphs focusing on the 2nd and 3rd quartiles to show changes in the bulk of the data distributions, thus forcing the x-axis between -1 and 1. We have now plotted the values considering the entire extent of the values. In addition to this, we have calculated the p-values (Wilcoxon test) of the comparisons between PTs and IBTRs for cell cycle and DNA repair transcript/proteins. The results of these tests were all significant and we have added them to the amended panels.

- On this point but as a separate concern, is the relevance of the observed difference in signature contribution e.g. signature 3 and 9. The boxplots already show a substantial overlap and I am still very skeptical about the biological importance/relevance of this difference. The increase in signature 9 is (estimated by the figure) something around 0.1 on average. I have a hard time believing that this is relevant and not the result of random chance. I was trying to understand what the signatures actually contain and the reference used in the manuscript (from 2013) only lists 21 signatures, not the 30 investigated here. There are 4 signature types listed on COSMIC and a quick search did not allow me to figure out which set of signatures was used in this study. My reason for checking was to understand if e.g. signature 3 is already correlated to e.g. ER or PgR status. If signature 3 is in fact the same as the one postulated in the corresponding reference, prior work (<https://www.ncbi.nlm.nih.gov/pmc/articles/PMC2880433/>) already suggested that tumors with BRCA1/2 mutations are typically ER negative and PgR negative.

We apologize for the confusion. We recognize the fact that mutational signatures are not a finalized set and are continuously expanded, as the reviewer notes. The mutational signatures that we investigated here were derived from the package MutationalPatterns (v3.3.0) and comprised the 30 mutational signatures available at the time of analysis (i.e. Mutational Signatures v2; https://cancer.sanger.ac.uk/signatures/signatures_v2/). We have clarified this in the Methods section (p 31, lines 584-5).

With regard to the small contribution, or contribution change as depicted in Figure 2, mutational signature contributions were reported as fractions (from 0 to 1), hence the seemingly small effect. We recognize that this was not made clear and have amended the

Methods section (p 31, lines 585-6). In addition to this we have amended related Figures, Supplementary Figures, and their legends.

With regards to BRCA1/2 mutations, the reviewer is correct in reporting the frequency of these changes in ER/PgR negative tumors. In our set only one sample expressed a coding BRCA1 mutation, which was acquired at the IBTR level. For this, the assumption that BRCA1/2 mutations are associated with signature 3 contribution does not hold true in our dataset.

- With respect to signature 17: While the association analysis was removed, the fact that it remains (according to the figure, being the first listed there) the top signature with contribution changes although only 2 samples showed a difference, raises concerns about the ranking procedure and again the biological importance of the observed changes.

We ranked the change in mutational signature contribution between PT-IBTR pairs by the absolute value of the mean change. This resulted in signatures 5, 3, 1, 9, and 17 to be the ones with the biggest change, respectively. For this, we do not believe that our selection method is wrong, just impacted by changes in few samples as in the case of signature 17. We have re-plotted Figure 2A following Reviewer 2 comments and plotted the top 10 (top 5 highest + top 5 lowest delta), where it shows clearly that signature 17 has a big contribution change in only two samples. On top of discussing this in the text, we have removed signature 17 from Figure 2B, where now signatures are ranked by absolute mean contribution change.

This is further corroborated by the observation that a surprisingly large number of statistical tests performed in this study only result in “borderline” (terminology used in the manuscript when referring to C>T transitions with a p-value ~0.076) significance. This loops back to the signature 3, where a p-value of 0.074 testing the relation to ER expression is referred to as “significant”. Examples for this are e.g. Fig S8 with 0 or 1 out of 9 tests and Fig 4 with 2 or 4 out of the 9 tests being significantly different when using a 0.01 or 0.05 cutoff, respectively.

We apologize for the confusion. We have specified in the methods the p-value cutoff for significance calling (p 36, line 693). In addition to this, we limit ourselves to report the p-values and do not make any claim with regard to the significance of the association between signature 3 change in contribution and ER status. While we have discussed the low contribution of mutational signatures and the limited power of our sample set in defining relationships with clinical variables in the Discussion section (p 24, lines 454-5), we do not think that borderline associations should be overlooked and discarded because they might constitute a query for a follow-up study including a larger set of samples.

Minor issues:

- Figure 2: There is a typo, referencing a panel D in the legend which does not exist (I assume it was meant to refer to new C). The legend still refers to the top 10 signatures in panel B although only 5 are shown.

We have amended the Figure legend.

- Figure 4: In the RNA clustering, a bit right of the center, two samples are labeled “matched” but belong to different groups. This must be a mistake in plotting/labeling matched pairs as the leafs of a dendrogram can be re-ordered without changing the clustering and thus the currently matched pair would not be plotted side-by-side anymore.

We noticed a typo in our script and have amended this error.

- The authors have addressed some of my concerns with respect to the complexity of the figures. However, I think this can still be further improved by e.g. not showing details which are not discussed in the manuscript. Most of the heatmaps contain annotations that are never discussed or even mentioned.

We have simplified all heatmaps presented in the manuscript.

- The cosine dissimilarity measure used, particularly in Figure S6 C and B, suggests that the authors have applied the cosine similarity on scaled expression data, otherwise a value of 1.5 (essentially $1 - (-0.5)$) is hard to explain. This may potentially explain the bi-modal distribution observed for RNA and protein expression data.

The cosine similarity for all data layers ranged from 0 to 1. We have amended Figure S6.

Reviewer #2 (Remarks to the Author):

The authors greatly improved the write-up of the manuscript. They comprehensively addressed the reviewers' comments, to the extent possible, and refocused on the research question asking why ipsilateral recurrent breast cancer may occur. They truly made an effort to improve the write-up of the manuscript.

Two minor comments:

Abstract, line 141, is genetic variants the right term or should it be somatic mutations?

We have amended the Abstract text.

Figure 2A, the heatmap looks like a white space - nothing in it. The authors may choose a better color coding to highlight the findings in this graph.

The reviewer points out an important notion, which relates to some signatures not displaying any change in contribution, or any contribution at all. This in turn resulted in several signatures being dropped for follow-up association analyses. To focus on only relevant signatures, we have re-plotted Figure 2A including the top 10 signatures with the biggest changes (i.e. top 5 highest delta + top 5 lowest delta).

REVIEWERS' COMMENTS:

Reviewer #2 (Remarks to the Author):

Thank you for the corrections to the manuscript.

Minor: line 919, "between" instead of "beyween".

Reviewer #3 (Remarks to the Author):

Review of rebuttal to Reviewer#1 concerns:

Major #1. Authors have addressed majority of the concern. It would be important to mention which wilcoxon test was used. Wilcoxon signed-rank test is more suitable for paired PT/IBTR analysis.

Major #2 Authors have addressed the concern substantially

Major #3 To increase comprehensibility of the results, authors should include the proposed aetiology for mutational signatures (As listed on COSMIC website) in Fig 2A.

Major #4 Authors have made changes to address the concern

Reviewer #2 (Remarks to the Author):

Thank you for the corrections to the manuscript.

Minor: line 919, "between" instead of "beyween".

Amended.

Reviewer #3 (Remarks to the Author):

Review of rebuttal to Reviewer#1 concerns:

Major #1. Authors have addressed majority of the concern. It would be important to mention which wilcoxon test was used. Wilcoxon signed-rank test is more suitable for paired PT/IBTR analysis.

We have specified which Wilcoxon test was employed in our analyses in our Methods section (p 35, line 678).

Major #2 Authors have addressed the concern substantially

n/a

Major #3 To increase comprehensibility of the results, authors should include the proposed aetiology for mutational signatures (As listed on COSMIC website) in Fig 2A.

We have added a legend with the proposed aetiology of each signature portrayed in Figure 2A.

Major #4 Authors have made changes to address the concern

n/a